# Reconsidering Degeneration of Token Embeddings with Definitions

## Abstract

While learning token embeddings via language modeling and weight tying remains the dominant paradigm, embeddings often degenerate into anisotropic (i.e., non-uniform) distributions in geometry space, limiting their expressiveness. This study first analyzes the fine-tuning dynamics of encoder-based pretrained language models (PLMs) and shows that their embeddings can largely preserve their geometric structure to defend against degeneration during fine-tuning. However, pretrained embeddings still suffer from anisotropic distribution, and low-frequency tokens tend to lose their semantics. To address this issue, we propose DefinitionEMB, a method that leverages lexical definitions to inject explicit semantics into embeddings while anchoring them to the pretrained geometric manifold to preserve PLMs' established geometric knowledge. Extensive experiments demonstrate the effectiveness of leveraging Wiktionary definitions on four PLMs: RoBERTa-base, BART-large, T5-large, and T5Gemma-l-ul2 across natural language understanding and abstractive text summarization.

## 1 Introduction

Token embeddings are the foundational inputs of language models, encoding lexical information into continuous vectors for downstream tasks (Parsing, 2009; Turian et al., 2010). Modern LMs typically learn these embeddings through language modeling objectives and optimize them jointly with model parameters (Mikolov et al., 2013), often with weight tying to reduce computation (Press & Wolf, 2017). Ideally, embeddings should efficiently utilize the vector space and maintain distinct semantic neighborhoods (Cai et al., 2021). However, evidence shows that token representations tend to degrade during language modeling: their spatial distribution becomes less diverse (**anisotropy**), and tokens with similar frequencies grow closer even when semantically unrelated. This degeneration harms low-frequency tokens most, particularly the rare subword tokens that dominate the vocabularies of PLMs, rapidly losing their expressiveness and becoming hard to differentiate. For example, consider a model summarizing the sentence: "102 people were injured in the accident..." The subword token for "02" is rare, and its input embedding is clustered indistinguishably with other rare numerals, such as "0000". As a result, the model may incorrectly output "10000 people were injured". This turns a moderate incident into a catastrophe, simply because the model could not tell the difference between two rare subwords. This critical challenge motivates our study on diagnosing and reconstructing the embeddings of rare subwords in PLMs.

To provide an intuitive illustration of this degeneration phenomenon, we train BART-large (Lewis et al., 2020) from scratch and visualize its token embeddings in 2D using SVD (Figure 1). At the beginning (Figure 1a), randomly initialized embeddings are uniformly distributed regardless of their semantics. After training (Figure 1b), embeddings degenerate into a narrow cone, indicating a global collapse of directional diversity. Tokens with similar frequencies cluster together regardless of meaning, revealing local degeneration where semantic distinctions vanish. While previous methods have been proposed to address token degeneration either by post-processing word embeddings or modifying model optimization strategies (Mu et al., 2018; Gong et al., 2018; Yu et al., 2022), these approaches primarily target word-level embeddings or models trained from scratch, and thus cannot be directly applied to real-world PLMs. Crucially, PLM vocabularies are dominated by low-frequency subword tokens (e.g., "iment" in "sediment"), which lack explicit lexical grounding, making their degeneration patterns and recovery strategies fundamentally different. Consequently, how degeneration

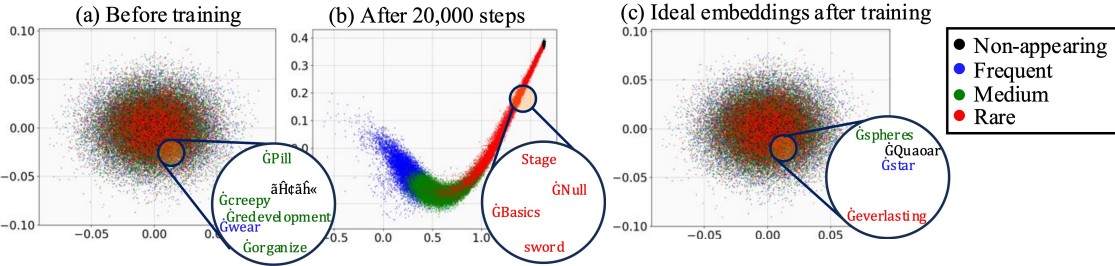

Figure 1: Training from scratch on CNN/DailyMail (CNNDM). (a) Initialized token embeddings are randomly distributed. (b) After 20,000 training steps, token embeddings degenerate into a narrow cone. (c) Ideally distributed embeddings group semantic related tokens.

unfolds in widely deployed PLMs and how it affects the representations of these critical rare subword tokens remain insufficiently understood.

To bridge this crucial gap, we conduct a systematic analysis of encoder-based PLMs (Section 4), which constitute a major family of models for natural language understanding and generation tasks. Our findings show that PLMs largely preserve their global embedding geometry during fine-tuning (i.e., *geometry-preserving*), but their pretrained rare subword embeddings still suffer from anisotropy and degraded semantic relatedness. While anisotropy is widely interpreted as the primary cause of embedding degeneration, our diagnostic results reveal that improving isotropy alone does not restore rare-token semantics and can even harm semantic quality during fine-tuning. These findings suggest that anisotropy is a secondary phenomenon and the core challenge lies in improving the semantic quality of rare subword tokens. Motivated by these findings, we ask the following question:

> *For subword embeddings in PLMs, how can we reconstruct them to achieve and maintain semantic relatedness during fine-tuning?*

To answer this research question, we propose DefinitionEMB, a method for reconstructing subword embeddings for PLMs in an architecture-agnostic manner. In DefinitionEMB, token embeddings are reconstructed using dictionary definitions as the primary source of lexical semantics. To safeguard these semantics during fine-tuning, we leverage the geometry-preserving property observed in PLMs. We thus introduce a mimicking objective as an $\ell_2$ regression loss (Pinter et al., 2017) to ensure compatibility with PLMs. In this setup, the pretrained vectors serve as geometric anchors; by aligning with them, a definition-derived embeddings acquire the capability to maintain their global distribution, thereby defending against degeneration during subsequent fine-tuning. Unlike the lightweight post-processing method DelDirection (Mu et al., 2018), we employ a reconstruction framework that requires a definition corpus for training to achieve these benefits, trading off computational efficiency for guaranteed semantic preservation. Building on prior findings of anisotropy in hidden representations (Ethayarajh, 2019; Cai et al., 2021) and in word-level embeddings (Gao et al., 2019), our contributions are as follows:

- **Insights into PLMs' Token Embedding Degeneration.** We first confirm that the input token embeddings of encoder-based PLMs are also anisotropic, with rare tokens suffer from semantic unrelatedness. Based on this, we provide two new observations regarding fine-tuning dynamics: (i) The relative geometry of token embeddings is largely preserved during fine-tuning, meaning that fine-tuning does not fix the semantic unrelatedness for rare tokens. (ii) Directly improving isotropy does not help recover semantic meaning. Instead, it introduces geometric fragility during fine-tuning."

- **Degeneration-Resistant Embeddings.** To the best of our knowledge, DefinitionEMB is the first method to reconstruct token embeddings for PLMs that simultaneously achieve *semantic relatedness* and *geometry-preserving*.

- **Empirical Validation.** We show the effectiveness of DefinitionEMB through extensive experiments on two natural language understanding and four text summarization datasets, covering four PLMs: RoBERTa-

base (Liu et al., 2019b), BART-large, T5Gemma-l-ul2 (Zhang et al., 2025), and T5-large (Raffel et al., 2020). A probing analysis further shows that the token semantics introduced by DefinitionEMB are preserved throughout the network and remain accessible at the final layer.

## 2 Related Work

**Learning token embeddings** refers to optimizing embeddings through language modeling, where a model is trained to predict the next token given its preceding or surrounding context, and token embeddings are updated via backpropagation. This setting requires a large amount of training data so that embeddings can acquire semantics from the contexts in which they appear (Khodak et al., 2018; Liu et al., 2019a; Devlin et al., 2019). However, rare tokens occur in few contexts, prompting methods that use lexical definitions as an alternative. For example, Tissier et al. (2017) align word-level embeddings using definitions, while others (Bahdanau et al., 2018; Zhang et al., 2021; Ruzzetti et al., 2022) directly construct word-level embeddings from definitions. SentenceBERT (Reimers & Gurevych, 2019) is designed to generate a single vector for an entire sentence. In contrast, modern PLMs use subword vocabularies where most subwords lack individual definitions, making it impossible to obtain their embeddings directly from a dictionary. To bridge this gap, our DefinitionEMB uses the full-word definition as input and decomposes its semantic information into multiple embeddings, one for each constituent subword. This ensures that the generated embeddings are consistent with the PLM's subword-level architecture.

**Embedding geometry** encodes token semantics through distances in the embedding space (Mimno & Thompson, 2017). Prior work has explored improving embedding geometry from multiple perspectives. Some methods enhance *semantic relatedness* by reducing frequency-induced disparities among word embeddings (Gong et al., 2018), while others preserve geometry during training by regulating gradient updates for rare tokens (Yu et al., 2022). Additional studies target *isotropy* through distance regularization (Zhang et al., 2020) or by removing dominant common components (Rajaee & Pilehvar, 2021; Biś et al., 2021). More recently, Sajjad et al. (2022) systematically analyzed post-processing methods for contextualized embeddings and showed that normalization and component removal can improve isotropy and lexical task performance, further confirming the anisotropic nature of PLM representations. However, these methods mainly focus on post-hoc geometric adjustments of contextualized embeddings and are not designed to introduce new token-level semantic supervision or to study rare subword degeneration under fine-tuning. In contrast, DefinitionEMB reconstructs token embeddings by injecting definitional semantics while anchoring them to the pretrained geometric manifold, simultaneously improving semantic relatedness and geometry preservation.

**Other representations in PLMs** have also been extensively studied, including token-level extensions and analyses of contextualized representations. Prior work like Chen et al. (2022) and Liang et al. (2023) focuses on extending BERT embeddings for out-of-vocabulary words, and Ethayarajh (2019) investigates the geometry of contextualized representations. More recently, Durrani et al. (2022) studies how the latent space of pretrained language models evolves during fine-tuning, revealing substantial layer-wise structural changes and task-specific reorganization. These methods provide valuable insights but are not directly aligned with our objective. Our focus is on understanding subword-level degeneration during fine-tuning, which exhibits failure patterns distinct from contextual geometry analyses. In particular, these studies do not examine the semantic unrelatedness and isotropy imbalance of rare subword tokens within PLM vocabularies, nor do they analyze how token-level geometric structures evolve under fine-tuning. These issues are central to the motivation of DefinitionEMB and remain unsolved in prior research.

## 3 Preliminaries

**Isotropy.** Let $\mathcal{V} = \{v_n\}_{n=1}^{|\mathcal{V}|}$ denote the predefined restricted vocabulary and $\mathbf{E} \in \mathbb{R}^{|\mathcal{V}| \times h_e}$ the pretrained token embedding matrix, where $h_e$ is the embedding dimension. The geometry of $\mathbf{E}$ is generally assumed to reflect linguistic regularities, *i.e.,* semantically similar tokens should have similar embeddings. To maximize the expressive power of $\mathbf{E}$, researchers often aim for high isotropy (*i.e.,* uniform distribution). To quantify this, Mu et al. (2018) proposed the isotropy score $I(\mathbf{E}) \in [0, 1]$, where a value closer to 1 indicates higher isotropy.

**Weight Tying.** Weight tying enforces equality between the input embedding matrix $\mathbf{E}$ and the output projection matrix $\mathbf{W}_{\text{out}}$, *i.e.,* $\mathbf{W}_{\text{out}} = \mathbf{E}$, which reduces parameters and aligns input-output representations, but incurs an undesired coupling during training, particularly affecting rare tokens. In generation tasks, the output probability of a token $\boldsymbol{w}_{t+1}$ is computed based on the hidden state $\boldsymbol{h}_t$ derived from the preceding context $\boldsymbol{w}_{<t}$ as follows:

$$P(\boldsymbol{w}_{t+1}|\boldsymbol{w}_{<t}) = \text{Softmax}(\mathbf{E} \cdot \boldsymbol{h}_t), \tag{1}$$

$$P(\boldsymbol{w}_{t+1} = v_i|\boldsymbol{w}_{<t}) = \frac{\exp(\boldsymbol{e}_i \cdot \boldsymbol{h}_t)}{\sum_{j=1}^{|\mathcal{V}|} \exp(\boldsymbol{e}_j \cdot \boldsymbol{h}_t)}, \tag{2}$$

where $\boldsymbol{e}_i$ denotes the embedding of the $i$-th token. During backpropagation, the loss function updates $\mathbf{E}$, not only to optimize output prediction but also to refine input embeddings. This coupling leads to: (1) Popular tokens dominate gradient updates, and tokens that co-occur in the same contexts are pulled closer together in embedding space to have similar embeddings; (2) Rare tokens receive sparse or noisy updates and are pushed in the opposite direction (Demeter et al., 2020). Over time, embeddings drift away based on token frequency, ultimately collapsing into a narrow cone, losing isotropy, and failing to preserve semantic relationships, a phenomenon known as embedding degeneration (Ethayarajh, 2019).

**DelDirection.** We adopt DelDirection (Mu et al., 2018) as our preliminary baseline for this study[1], as it is a post-processing method applicable to subword-level vocabularies for improving embedding isotropy. DelDirection assumes that all token representations share a common mean vector $\boldsymbol{\mu}$ and are dominated by a set of principal directions. These shared components affect the isotropy of token embeddings. To address this, DelDirection proposes removing both the mean vector and principal directions to improve isotropy. The procedure for any token $\boldsymbol{e} \in \mathbf{E}$ can be formulated as follows:

$$\bar{\boldsymbol{e}} = (\boldsymbol{e} - \boldsymbol{\mu}) - \sum_{i=1}^{D} (\boldsymbol{u}_i^\top \boldsymbol{e}) \boldsymbol{u}_i, \tag{3}$$

where $\{\boldsymbol{u}_1, \dots, \boldsymbol{u}_D\} = \text{PCA}(\mathbf{E} - \mathbf{1}\boldsymbol{\mu}^\top)$ represents the top $D$ principal components via PCA, and $\mathbf{1} \in \mathbb{R}^{|\mathcal{V}|}$ is an all-ones vector. In their study, they set $D = h_e/100$. While this equation removes the dominant directions to improve isotropy, it fundamentally disregards the underlying semantic relationships between tokens. Before introducing our method, we present a diagnostic pre-experimental investigation in Section 4. The purpose of this investigation is not to compare baselines but to reveal the geometric and semantic degeneration issues of PLMs and DelDirection during fine-tuning, which motivate the design of DefinitionEMB.

## 4   Pre-Experimental Investigation

Section 4 serves as a pre-experimental study aimed at identifying three core limitations: semantic unrelatedness, isotropy imbalance, and geometric fragility. While anisotropic embedding geometry and narrow-cone effects have been well documented at the word or contextual representation level (Sajjad et al., 2022; Ethayarajh, 2019; Durrani et al., 2022), our investigation extends these findings to the subword embedding space that PLMs directly operate on. We show that similar geometric degradation emerges at the subword level and disproportionately affects rare tokens during fine-tuning, providing direct motivation for targeted rare-token reconstruction. Rather than evaluating baselines, this section extracts empirical findings that motivate the design requirements of DefinitionEMB in Section 5. The full baselines for downstream evaluation will be introduced in Section 6.

**Pre-Experimental Setup:** We conduct all analyses in this section using BART-large and T5-large fine-tuned on the CNNDM dataset. Tokens are projected into 2D using PCA for visualization. Tokens are grouped into frequent/medium/rare based on their corpus frequency (30%/50%/20%). These settings are used solely for diagnostic visualization, not for evaluating downstream task performance.

**DelDirection as a Diagnostic Example.** As a representative post-processing method, DelDirection shows how removing principal directions improves isotropy but disrupts the original geometric structure. We

---

[1]DelDirection is our operational shorthand for the All-but-the-Top method introduced by Mu et al. (2018).

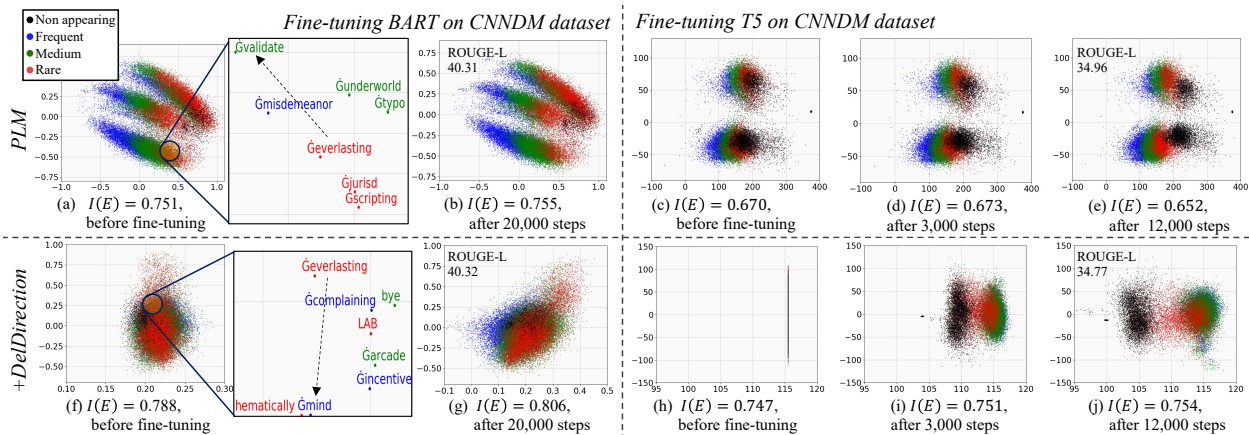

Figure 2: Token embeddings projected onto the first two singular vectors. **Top**: During fine-tuning, PLMs' embeddings across frequency groups maintain relative positions, showing *geometry-preserving*. **Bottom**: DelDirection is *geometric fragility*, lower-frequency tokens drift farther from original positions. In **(a)** and **(f)**, the rare token "Ġeverlasting" has only one semantic related neighbor, marked by the dashed line.

include DelDirection here not as a baseline for comparison, but as a diagnostic example to examine how geometry-oriented post-processing interacts with PLM subword embeddings during fine-tuning. We apply the official implementation using the same hyperparameters.

Then, as shown in Figure 2, we fine-tune PLMs and obtain the following three main findings:

[**Finding-1**]. *PLMs exhibit anisotropy and semantic degeneration among rare subword tokens.* Prior work (Ethayarajh, 2019; Mu et al., 2018) has shown that contextual representations tend to be anisotropic. Our diagnostic analysis extends this observation to subword-level pretrained token embeddings during fine-tuning, which has not been systematically examined. In Figure 2 (a,c), PLM embeddings show isotropy values between 0.670 and 0.751 and form a narrow cone structure dominated by high-frequency tokens before fine-tuning. Rare tokens (highlighted in red) collapse into a tight directional subspace, indicating low geometric diversity. To further examine semantic degeneration, we inspect the nearest neighbors of a sampled rare token such as "Ġeverlasting," whose neighbors are semantically unrelated, demonstrating that rare tokens fail to form coherent semantic neighborhoods. These empirical patterns directly motivate the need for explicit semantic reconstruction.

[**Finding-2**]. *PLMs are geometry-preserving during fine-tuning.* Figures 2 (b,e) show that, after fine-tuning, the overall topology of the embedding space remains largely unchanged for both BART and T5. The principal axes, token-frequency ordering, and relative spatial arrangement of token clusters are preserved, indicating that fine-tuning does not distort the global geometry learned during pretraining. This geometry-preserving behavior is consistent across architectures, as RoBERTa-base exhibits similar stability (Appendix C.1). These observations motivate designing a method that respects the original PLM geometry while improving rare-token semantics and isotropy.

[**Finding-3**]. *DelDirection can improve isotropy but suffers from geometric fragility and semantic unrelatedness.* As a representative post-processing method, DelDirection removes top principal directions. Although this increases isotropy, Figures 2 (g,i,j) show that the resulting embedding space becomes substantially distorted: unlike the pretrained embeddings, the cluster topology in DelDirection collapses, with token groups of different frequencies shifting at varying speeds during fine-tuning. This inconsistent migration indicates a clear geometric fragility in DelDirection. Furthermore, in Figure 2 (f), the rare token "Ġeverlasting" still fails to form a coherent semantic neighborhood with only one meaningful neighbor. This suggests that while DelDirection modifies global geometry, it does not provide semantic reconstruction for rare tokens and therefore does not alleviate semantic unrelatedness. These negative results challenge the common belief that anisotropy is the primary cause of degeneration, and instead highlight semantic deficiency as a key limitation that must be addressed to improve rare-token representations.

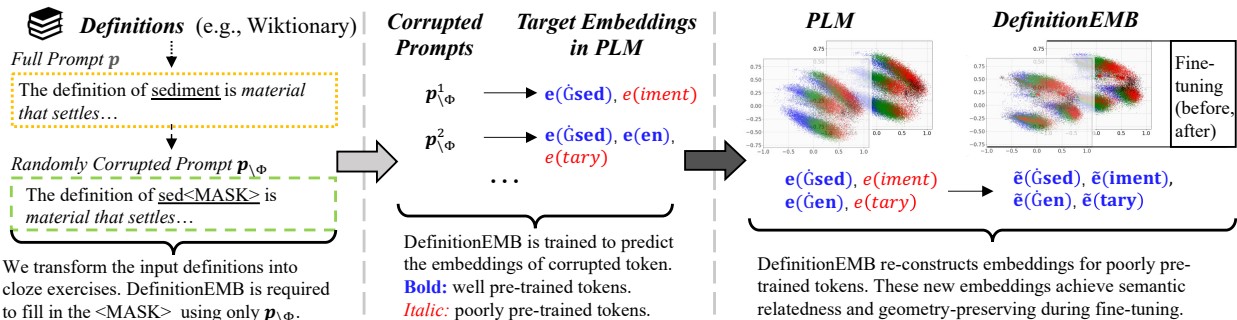

Figure 3: Overview of constructing definition embeddings to replace pretrained embeddings. The prefix $\dot{G}$ indicates that a subword is at the beginning of a word in the BART/RoBERTa BPE tokenizer.

## 5 Methodology

**Overview.** In Figure 3, motivated by the observation that improving isotropy alone does not yield semantic benefits, DefinitionEMB is designed to enhance rare-token semantic relatedness while preserving the pretrained geometric structure during fine-tuning. DefinitionEMB is a definition-driven method that contrasts with post-processing methods. It employs a denoising autoencoder and requires external dictionary definitions to achieve semantic reconstruction for subwords.

### 5.1 Reconstruction for Semantic Relatedness

PLMs based on the language modeling objective typically derive token semantics from surrounding contexts. However, for rare tokens, diverse and high-quality contexts are inherently difficult to retrieve; consequently, contextual signals are often noisy or insufficient to convey precise lexical meanings. According to Ausubel (1968), linking a new word to known relevant concepts fosters a deep understanding of that word. For example, learning the word "sediment" can be guided by its lexical definition "material that settles to the bottom of a liquid". This suggests that dictionary definitions provide a "semantic shortcut"—explicit, dense, and token-specific semantic supervision that cannot be inferred from sparse co-occurrence patterns alone. Motivated by this insight, we propose DefinitionEMB, a definition-driven learner that leverages definitional semantic signals to directly enrich rare-token representations within the pretrained manifold.

A cloze exercise is a task where a word in a sentence is replaced with a placeholder (e.g., a <MASK> token), and the model is required to predict the replaced word from the surrounding context. Our training adopts this paradigm in the form of a denoising autoencoder (Vincent et al., 2010). As shown in Figure 3 (left), we first construct the full prompt $\boldsymbol{p}$ by combining a word $\boldsymbol{c}$ with its definition. To improve coverage, we include lexical variants of $\boldsymbol{c}$, such as its original form, lowercase form, and capitalized form. Then, to prevent over-reliance on the input text and to encourage generalizable representation learning, we randomly corrupt a subset of tokens $\Phi$ of the tokens of $\boldsymbol{c}$ in the full prompt $\boldsymbol{p}$, replacing them with the mask token <MASK>, and obtain a corrupted prompt $\boldsymbol{p}_{\backslash\Phi}$. Finally, DefinitionEMB reconstructs the embeddings of the masked tokens based on the definition context. Specifically, if the $k$-th token in the full prompt $\boldsymbol{p}$ is masked (denoted by $\boldsymbol{p}_k \in \Phi$), DefinitionEMB reconstructs its semantic related definition embedding $\tilde{\mathbf{e}}(\boldsymbol{p}_k) \in \mathbb{R}^{h_e}$ as follows:

$$
\begin{aligned}
\boldsymbol{h}_{g(k)} &= f(k, \boldsymbol{p}_{\backslash\Phi}; \theta^{\mathrm{PT}}), \\
\tilde{\mathbf{e}}(\boldsymbol{p}_k) &= \mathbf{W} \cdot \boldsymbol{h}_{g(k)},
\end{aligned} \tag{4}
$$

where $g(\cdot)$ maps a corrupted position to its prediction position; specifically, for an encoder-only PLM, $g(k) = k$; for an encoder-decoder PLM, $g(k)$ refers to the position of the token preceding $\boldsymbol{p}_k$ in the decoder. $\boldsymbol{h}_{g(k)} \in \mathbb{R}^{h_s}$ is the last hidden state at position $g(k)$. $\mathbf{W} \in \mathbb{R}^{h_e \times h_s}$ is a projection matrix that transforms the semantic representation of the input definition into the embedding of the token $\boldsymbol{p}_k$. Here, $f(\theta^{\mathrm{PT}})$ indicates that DefinitionEMB is initialized from the given PLM, allowing it to leverage pretrained parameters.

---

(a) **Full prompt with word definition**

The definition of discomfort is To cause annoyance or distress to. Its part-of-speech, bpe-form without space, capitalization, and uppercase are verb, discomfort, Discomfort, and DISCOMFORT.

(b) **Source for the encoder-only PLM**

The definition of <MASK> is To cause annoyance or distress to. Its part-of-speech, bpe-form without space, capitalization, and uppercase are verb, <MASK>comfort, ĠDis<MASK>, and <MASK> COM *BS ĠNations*.

(c) **Target for the encoder-only PLM**

The definition of discomfort is To cause annoyance or distress to. Its part-of-speech, bpe-form without space, capitalization, and uppercase are verb, discomfort, Discomfort, and DISCOMFORT.

(d) **Source for the encoder-decoder PLM**

The definition of $<MASK_1>$ is To cause annoyance or distress to. Its part-of-speech, bpe-form without space, capitalization, and uppercase are verb, $<MASK_2>$, $<MASK_3>$, and $<MASK_4>$.

(e) **Target for the encoder-decoder PLM**

$<MASK_1>$ discomfort $<MASK_2>$ discomfort $<MASK_3>$ Discomfort $<MASK_4>$ DISCOMFORT

Figure 4: Example of constructing prompts for the word "discomfort". $<MASK_i>$ denotes the $i^{\text{th}}$ mask token. *Italic* indicates randomly replaced tokens.

In Figure 4, depending on different PLM architectures, we design two masking strategies: (1) a BERT-style strategy (Devlin et al., 2019) for encoder-only PLMs (Figure 4 (b-c)). For tokens corresponding to lexical variants of $c$ in $p$, we randomly mask them during training by 50% replaced with <MASK>, 25% replaced by random tokens, and 25% left unchanged.[2] During inference, we mask only one token at a time to generate its embedding deterministically. This corrupted prompt is then input to the encoder, and $\boldsymbol{h}_{g(k)}$ is computed at position $k$ for the masked $\boldsymbol{p}_k$; and (2) a T5-style strategy (Raffel et al., 2020) for encoder-decoder PLMs (Figure 4 (d-e)). Let $\gamma$ denote the positions in $p$ where variants of $c$ appear, ordered by their occurrence. To construct the input sentence, we replace each variant $\boldsymbol{c}_j$ (at position $\gamma_j$) with a sentinel token $<MASK_j>$, producing a corrupted prompt for the encoder. The target sequence $\boldsymbol{y}$ is then formed by interleaving the mask tokens and their corresponding variants: $\boldsymbol{y} = (<MASK_1>, \boldsymbol{w}_1, \ldots, <MASK_{|\gamma|}>, \boldsymbol{w}_{|\gamma|})$. To reconstruct the embedding of a token $\boldsymbol{p}_k$ within $\boldsymbol{c}_j$, we feed the target prefix up to $\boldsymbol{p}_k$ into the decoder. The hidden state $\boldsymbol{h}_{g(k)}$ is extracted in the final position of the decoder. Then, the model is encouraged to fully understand and utilize the semantic content of the definition to recover the missing information. This training strategy enables DefinitionEMB to learn robust and generalizable token embeddings, even in the presence of noisy or incomplete input.

## 5.2 Reconstruction for Geometry-Preserving

While external definitions help incorporate explicit semantics into the embeddings, a crucial issue is ensuring these semantics persist without degrading during subsequent fine-tuning. Our analysis in § 4 reveals a geometry-preserving property of PLMs, whereby pretrained embeddings maintain their distribution during fine-tuning. We thus reconstruct embeddings to inherit such stability, thereby defending against degeneration. DefinitionEMB is trained using a mimicking objective (Pinter et al., 2017), which optimizes embeddings by matching definition-driven embeddings to well-pretrained ones. The pretrained embeddings thus serve as geometric anchors. Specifically, we hypothesize that DefinitionEMB can effectively encode definitional semantics due to Eq. (4). Consequently, tokens with similar definitions, regardless of their frequencies, should be mapped to similar embeddings and cluster in the same regions. Assuming well-pretrained embeddings are ideally distributed, mimicking them enables rare tokens to be integrated into the structure originally dominated by these well-pretrained ones, rather than forming isolated and anisotropic clusters. The resulting embeddings thus inherit the geometry-preserving property.

---

[2]Pre-experiments showed that a large <MASK> ratio results in slow convergence, while a small ratio causes limited change between $\mathbf{e}(\boldsymbol{p}_k^m)$ and $\tilde{\mathbf{e}}(\boldsymbol{p}_k^m)$.

Formally, let $\mathcal{D} = \{(\boldsymbol{c}^m, \boldsymbol{p}^m)\}_{m=1}^M$ denote a corpus of $M$ word-definition pairs. For the $k$-th token in $\boldsymbol{p}^m$, DefinitionEMB is optimized by minimizing the mean squared error between the pretrained embedding $\mathbf{e}(\boldsymbol{p}_k^m)$ and its reconstructed definition embedding $\tilde{\mathbf{e}}(\boldsymbol{p}_k^m)$, formulated as follows:

$$\mathcal{L} = \sum_{m=1}^M \frac{\sum_{\boldsymbol{p}_k^m \in \Phi^m} \|\mathbf{e}(\boldsymbol{p}_k^m) - \tilde{\mathbf{e}}(\boldsymbol{p}_k^m)\|^2}{|\Phi^m|}, \tag{5}$$

where $|\Phi^m|$ is the number of masked tokens for $\boldsymbol{p}^m$. This objective enforces consistency of $\tilde{\mathbf{e}}(\boldsymbol{p}_k^m)$ for the same token across definitions and constrains reconstructed embeddings to stay close to the pretrained embedding manifold. To identify reliable anchors $\mathbf{e}(\boldsymbol{p}_k^m)$, we leverage the fact that large-scale pretraining yields a well-structured space for frequently occurring tokens. Recognizing that both popular and rare tokens may provide useful geometric information, following Burns et al. (2023), we incorporate definitions for the entire vocabulary $\mathcal{V}$. This allows the model to leverage consistent geometric patterns while naturally filtering out noise from unreliable anchors. In this design, semantic information comes from definitions, while the mimicking loss provides essential geometric regularization without propagating unreliable semantics from poorly trained rare-token embeddings.

### 5.3 Reconstruction for Downstream Tasks

Given a specific task, we first remove the tokens from $\mathcal{V}$ that do not appear in the task-specific fine-tuning dataset. The remaining tokens form a subset denoted by $\mathcal{V}_{[\text{task}]}$, which is sorted in descending order based on their frequency in the pretaining dataset. This ranking helps identify rare tokens that are less optimized during pretraining, which are considered the primary candidates for replacement.

To refine their representations and reduce potential noise, we replace a portion of these rare tokens with external embeddings obtained from definitions. Specifically, we replace the embeddings of the last $\min(\alpha\% \cdot |\mathcal{V}|, |\mathcal{V}_{[\text{task}]}|)$ tokens in the frequency-ranked list, where $\alpha$ is a tunable hyperparameter that controls the replacement ratio. For each selected token $v$, its original embedding $\mathbf{e}(v)$ is replaced with a definition-based embedding $\tilde{\mathbf{e}}(v)$ as $\mathbf{e}(v) \leftarrow \tilde{\mathbf{e}}(v)$. This replacement is performed before fine-tuning and aims to mitigate the impact of under-trained or low-quality embeddings associated with rare tokens. Crucially, the objective in Eq. (5) ensures that DefinitionEMB specifically improves representations where pretrained models struggle most. If a token's pretrained embedding is already high-quality and semantically dense, the discrepancy between the reconstructed and pretrained embedding will be minimal, even if its frequency is low in the downstream task. Consequently, our method naturally preserves well-learned semantics while focusing its "corrective" power on truly deficient rare-token embeddings.

## 6 Experiments

### 6.1 Experimental Settings

**Datasets.** Following Lewis et al. (2020), we evaluate DefinitionEMB on natural language understanding and text summarization, as shown in Table 1. For summarization, we adopt CNNDM (Hermann et al., 2015), Xsum (Narayan et al., 2018), BillSum (Kornilova & Eidelman, 2019) and Y-BIGPATENT (Sharma et al., 2019a) datasets. For natural language understanding, we use Microsoft Research Paraphrase Corpus (MRPC) (Dolan & Brockett, 2005), and Recognizing Textual Entailment (RTE) (Magnini, 2015).

**Baselines.** We select baselines that are designed to address anisotropic distribution, since PLMs are geometry-preserving. Thus, we adopted DelDirection (Mu et al., 2018) as our baseline, since other isotropy-oriented methods require additional semantic relationships between tokens, which are unavailable for subwords. Note that while DelDirection improves isotropy by altering the global geometric distribution, DefinitionEMB is specifically designed to preserve the pretrained embedding manifold. Consequently, these two approaches are not inherently complementary. Furthermore, we incorporate an additional baseline to apply z-score normalization after DelDirection, termed Del-zscore. This choice is based on the findings of Sajjad et al. (2022), which reported enhanced word similarity performance using this combined approach. We adopt the RoBERTa-base model as our backbone PLM with an encoder-only architecture. BART-large and T5-large

| Dataset | Number of Train | Number of Validation | Number of Test |
|---|---|---|---|
| BillSum | 17,054 | 1,895 | 3,269 |
| Y-BIGPATENT | 124,397 | 6,911 | 6,911 |
| XSum | 204,045 | 11,332 | 11,334 |
| CNNDM | 287,227 | 13,368 | 11,490 |
| RTE | 2,490 | 277 | 3,000 |
| MRPC | 3,668 | 408 | 1,725 |

Table 1: Detailed statistics of the datasets.

| Model | CNNDM | Y-BIGPATENT | XSUM | Billsum | MPRC | RTE |
|---|---|---|---|---|---|---|
| BART | 100 | 30 | 10 | 7 | 5 | 3 |

Table 2: Tuned Hyperparameter $\alpha$ for Experiments.

models served as backbone PLMs with an encoder-decoder architecture. Both RoBERTa-base and BART-large used the same vocabulary $\mathcal{V}$. To validate our encoder-based method on contemporary PLMs, we incorporate T5Gemma-l-ul2 (Zhang et al., 2025) as a crucial modern backbone. This choice bridges the gap with the prevailing decoder-only trend (e.g., Gemma2 (Team et al., 2024)) since T5Gemma adapts Gemma2 into a compatible encoder-decoder structure. For simplicity, we refer to this baseline as Gemma hereafter. Table 2 provides $\alpha$ for downstream tasks. More details of hyperparameters are listed in Appendix A.

**Implementation Details.** We used 1.5 GB of English definitions extracted from Wiktionary to train DefinitionEMB. These definitions were randomly divided into 1,454,327 training and 10,000 validation examples. We extracted 1,388 and 315 definitions corresponding to the BART and T5 vocabularies, respectively, from online sources. These were preliminary used to train embeddings for numerical and named entity tokens. In total, 1,455,715 and 1,454,642 examples were used to train DefinitionEMB based on the BART and T5 vocabularies, respectively. During inference, the same definitions were reused to load the corresponding definition embeddings. Once a token embedding is replaced, it is not replaced again during the remainder of the procedure. Tokens without a corresponding definition (*e.g.,* ")=(") were excluded from replacement. Excluding non-definable tokens, Table 3 summarizes the size of dictionary-grounded vocabulary across PLMs. Table 4 lists the runtime statistics for BART and T5, where model training are conducted on 4 NVIDIA A100 (80GB) GPUs. Although DefinitionEMB requires substantial pre-processing time, this is a one-time cost. Once trained, DefinitionEMB can be directly applied during inference with high efficiency.

| | BART / RoBERTa | T5 | Gemma |
|---|---|---|---|
| **Total Vocabulary Size** | 50,265 | 32,128 | 256,000 |
| **Grounded Vocabulary Size** | 47,960 | 25,994 | 128,542 |
| **Vocabulary Coverage (%)** | 95.4% | 80.9% | 50.2% |

Table 3: Coverage statistics of dictionary-grounded vocabulary across different PLMs.

**Reproducibility and Significance.** All results in this paper are averaged over three independent random seeds, and for experiments where repeated evaluation is feasible, we additionally report the standard deviation and statistical significance to assess the stability of improvements. Statistical significance is computed using paired bootstrap resampling (Koehn, 2004). Specifically, we resample the test set 1,000 iterations with replacement. Before resample, we aggregate the results from all seeds to form a combined test pool. This approach simultaneously accounts for variations in both the random seeds and the data samples. For GLUE test sets, significance testing is not applicable because evaluation is performed through a fixed online server. For benchmarks such as GLUE, CNNDM, and XSum, we utilize official train/validation/test splits. This is because k-fold cross-validation is generally not adopted in PLM research due to the high computational cost

| Model | Pre-training DefinitionEMB | Downstream Fine-tuning | Downstream Inference (Per Sample) |
|---|---|---|---|
| T5 | - | 12.1h | 5.2 sec |
| + DelDirection | - | 11.9h | 5.2 sec |
| + Del-zscore | - | 12.4h | 4.6 sec |
| + DefinitionEMB | 12.0h | 12.2h | 4.6 sec |
| BART | - | 9.6h | 3.6 sec |
| + DelDirection | - | 10.8h | 4.0 sec |
| + Del-zscore | - | 9.5h | 2.0 sec |
| + DefinitionEMB | 8.4h | 9.8h | 3.6 sec |

Table 4: Runtime statistics. The downstream tasks are evaluated on the CNNDM dataset.

of full-model fine-tuning and the need to maintain consistent rare-token statistics. This protocol ensures methodological comparability with prior work and reliable variance estimation.

## 6.2 Evaluation on Downstream Tasks

### 6.2.1 Performance on Rare Tokens

A core motivation of this work is to address the long-standing challenge that PLMs often struggle to understand or generate texts involving rare tokens due to the lack of semantic representations. To examine whether the reconstructed embeddings are compatible with the remaining parameters of the PLMs in mitigating this issue, we design the following three experiments.

| Model | MRPC | RTE |
|---|---|---|
| RoBERTa | $87.5_{\pm 0.3}$ | $73.9_{\pm 2.2}$ |
| + DelDefinition | $86.9_{\pm 0.4}$ (-0.6) | $72.3_{\pm 1.1}$ (-1.6) |
| + Del-zscore | $69.0_{\pm 1.8}$ (-18.5) | $50.3_{\pm 0.0}$ (-23.6) |
| + DefinitionEMB | $\mathbf{87.7}_{\pm 0.4}$ (+0.2) | $\mathbf{75.3}_{\pm 0.7}$ (+1.4) |
| BART | $87.8_{\pm 0.3}$ | $82.4_{\pm 1.0}$ |
| + DelDefinition | $87.3_{\pm 1.2}$ (-0.5) | $78.7_{\pm 4.3}$ (-3.7) |
| + Del-zscore | $74.8_{\pm 1.1}$ (-13.0) | $51.7_{\pm 1.3}$ (-30.7) |
| + DefinitionEMB | $\mathbf{88.3}_{\pm 0.3}$ (+0.5) | $\mathbf{83.3}_{\pm 0.6}$ (+0.9) |

Table 5: Classification accuracy (mean$_{\pm \text{std}}$ across 3 runs) on MRPC and RTE datasets.

**Understanding Sentences Containing Rare Tokens.** We evaluate DefinitionEMB on NLU tasks using MRPC and RTE, two small-scale datasets with fewer than 4,000 training samples. Because these datasets provide limited supervision, they primarily test whether performance improvements stem from the quality of reconstructed embeddings rather than task-specific adaptation. Table 5 reports our test results from the public leaderboard.[3] Both DelDirection and Del-zscore perform worse than the original backbone. This could imply that although removing principal components and normalization helps refine static representations (Sajjad et al., 2022), such geometric adjustments alone are insufficient for introducing the new semantics needed in downstream fine-tuning tasks. DefinitionEMB slightly improves average accuracy for both RoBERTa and BART, showing that the reconstructed embeddings may even assist general sentence-level reasoning. Since NLU benchmarks contain few rare tokens (<1%), the observed accuracy gain, though slight, directly reflects the efficacy of DefinitionEMB in selectively improving the representations of these low-frequency tokens.

**Generating Summarization Containing Rare Tokens.** We further evaluated DefinitionEMB on text summarization—a task requiring deep semantic understanding and coherent generation. To evaluate the ability of DefinitionEMB to handle rare tokens, we conducted experiments on 65/60/102 data pairs of the

---

[3]https://gluebenchmark.com/

| Model | ROUGE-1 ↑ | ROUGE-2 ↑ | ROUGE-L ↑ |
|---|---|---|---|
| BART | $34.99_{\pm0.57}$ | $14.67_{\pm0.44}$ | $32.19_{\pm0.57}$ |
| +DelDirection | $35.13_{\pm0.40}$ (+0.14) | $15.05_{\pm0.46}$ (+0.38) | $32.33_{\pm0.54}$ (+0.14) |
| +Del-zscore | $33.73_{\pm0.56}$ (-1.26) | $14.26_{\pm0.62}$ (-0.41) | $31.58_{\pm0.31}$ (-0.61) |
| +DefinitionEMB | $\mathbf{36.11}^{\ddagger}_{\pm0.24}$ **(+1.12)** | $\mathbf{15.75}^{\ddagger}_{\pm0.39}$ **(+1.08)** | $\mathbf{33.44}^{\ddagger}_{\pm0.13}$ **(+1.25)** |
| T5 | $36.81_{\pm1.29}$ | $16.50_{\pm0.93}$ | $30.77_{\pm1.67}$ |
| +DelDirection | $36.98_{\pm0.45}$ (+0.17) | $16.56_{\pm0.47}$ (+0.06) | $30.84_{\pm0.23}$ (+0.06) |
| +Del-zscore | $37.32_{\pm0.76}$ (+0.51) | $16.56_{\pm0.32}$ (+0.06) | $31.09_{\pm0.65}$ (+0.32) |
| +DefinitionEMB | $\mathbf{37.57}^{\ddagger}_{\pm0.39}$ **(+0.77)** | $\mathbf{16.65}_{\pm0.12}$ **(+0.15)** | $\mathbf{31.43}^{\ddagger}_{\pm0.45}$ **(+0.66)** |
| Gemma | $14.04_{\pm0.84}$ | $1.60_{\pm0.10}$ | $12.98_{\pm0.66}$ |
| +DelDirection | $13.44_{\pm0.21}$ (-0.60) | $1.50_{\pm0.07}$ (-0.10) | $12.57_{\pm0.21}$ (-0.41) |
| +Del-zscore | $11.17_{\pm2.33}$ (-2.87) | $\mathbf{1.80}_{\pm0.45}$ (+0.20) | $10.25_{\pm2.23}$ (-2.73) |
| +DefinitionEMB | $\mathbf{14.52}^{\ddagger}_{\pm0.55}$ **(+0.48)** | $1.60_{\pm0.04}$ (+0.00) | $\mathbf{13.39}_{\pm0.65}$ **(+0.41)** |

Table 6: Model performance (mean$_{\pm\text{std}}$ across 3 runs) on the CNNDM subset in generating rare token-contained summaries.

CNNDM test set for BART/T5/Gemma-related models. In this subset, each target summary consisted of a high proportion of rare tokens (at least 5%). Note that BART, T5, and Gemma rely on distinct subsets due to their different vocabularies; see Appendix B.2 for details. We evaluated model performance on these datasets using the ROUGE scores. Table 6 shows the results on the three PLMs. (1) DefinitionEMB improves all metrics by over 1.0 point compared to BART, while DelDirection shows marginal gains (+0.14 ROUGE1 and ROUGEL). (2) DefinitionEMB achieves substantial gains on T5 (e.g., +0.77 ROUGE-1), significantly larger than DelDirection's minor improvements (e.g., +0.17 ROUGE-L). (3) DefinitionEMB provides clear gains for Gemma (+0.48 ROUGE-1), whereas DelDirection degrades Gemma's performance. Furthermore, baselines relying solely on geometric operations show unstable performance across different backbones. Notably, Del-zscore significantly degrades Gemma's performance by -2.73 ROUGE-L points. To further investigate this disparity, Appendix C.2 confirms that DefinitionEMB improves isotropy, particularly for rare tokens. When viewed alongside the gains in Table 6, these results demonstrate that DefinitionEMB goes beyond mere isotropy improvement; rather, it introduces critical semantic information that post-hoc geometric operations alone cannot recover. These findings confirm that DefinitionEMB effectively strengthens the semantic representations of rare tokens rather than simply smoothing them toward frequent-token distributions.

| Model | BART | +DelDirection | +DefinitionEMB |
|---|---|---|---|
| **Accuracy** | $96.4_{\pm0.3}$ | $84.8_{\pm10.0}$ | $\mathbf{98.3}_{\pm1.0}$ |

Table 7: Semantic preservation evaluation in the final embedding layer (mean$_{\pm\text{std}}$ across 3 runs).

**Preserving Semantics in Deep Layers.** To investigate why DefinitionEMB facilitates understanding and generation of rare tokens, we hypothesize that its reconstructed information is not only encoded in the input embedding layer, but also persist through the model's deeper layers, enabling downstream usage. To evaluate this, we adopt a probing setup inspired by Allen-Zhu & Li (2024). For each CNNDM fine-tuned model, we froze the model and trained a classifier to classify the hidden states of the input words of the last layer as "numeric $< 1,000$", "numeric $> 1,000$", or "others". In inference, we evaluated accuracy on 78 rare numeric tokens and 78 rare non-numeric tokens. (See Appendix B.3 for details.) The overall accuracy is shown in Table 7. Using DefinitionEMB achieves the highest accuracy at 98.3%, indicating that the semantic signals introduced by DefinitionEMB are preserved throughout the network and remain accessible at the final layer, supporting improved semantic generation in downstream tasks. The improved recoverability of rare-token identities by deeper layers reflects that the definitional semantic signals propagate through the model, thereby outperforming pretrained contextual exposure alone. Taken together with our analyses in Section 6.2.1, these results confirm that DefinitionEMB successfully strengthens semantics for poorly pretrained embeddings.

These enhanced representations are aligned with the PLM parameters, thereby yielding substantial gains in downstream semantic tasks.

| Dataset | Model | ROUGE-1 | ROUGE-2 | ROUGE-L |
|---|---|---|---|---|
| **CNNDM** | BART | $43.57_{\pm0.27}$ | $20.93_{\pm0.20}$ | $40.31_{\pm0.26}$ |
| | +DelDirection | $43.59_{\pm0.30}$ $(+0.02)$ | $20.93_{\pm0.18}$ $(+0.00)$ | $40.32_{\pm0.29}$ $(+0.01)$ |
| | +Del-zscore | $42.64_{\pm0.26}$ $(-0.93)$ | $19.67_{\pm0.36}$ $(-1.26)$ | $39.39_{\pm0.24}$ $(-0.92)$ |
| | +DefinitionEMB | $\mathbf{43.78}^{\ddagger}_{\pm0.09}$ $(+0.21)$ | $\mathbf{20.94}_{\pm0.09}$ $(+0.01)$ | $\mathbf{40.52}^{\ddagger}_{\pm0.09}$ $(+0.21)$ |
| **Y-BIGPATENT** | BART | $43.96_{\pm0.19}$ | $18.92_{\pm0.20}$ | $37.80_{\pm0.18}$ |
| | +DelDirection | $43.91_{\pm0.11}$ $(-0.05)$ | $18.85_{\pm0.10}$ $(-0.07)$ | $37.79_{\pm0.10}$ $(-0.01)$ |
| | +Del-zscore | $34.38_{\pm7.30}$ $(-9.58)$ | $10.49_{\pm4.60}$ $(-8.43)$ | $29.71_{\pm5.47}$ $(-8.09)$ |
| | +DefinitionEMB | $\mathbf{44.16}^{\ddagger}_{\pm0.09}$ $(+0.20)$ | $\mathbf{19.06}^{\ddagger}_{\pm0.11}$ $(+0.14)$ | $\mathbf{38.01}^{\ddagger}_{\pm0.09}$ $(+0.21)$ |
| **XSum** | BART | $43.76_{\pm0.38}$ | $20.40_{\pm0.32}$ | $34.65_{\pm0.47}$ |
| | +DelDirection | $43.90_{\pm0.19}$ $(+0.14)$ | $20.58_{\pm0.15}$ $(+0.18)$ | $34.86_{\pm0.25}$ $(+0.21)$ |
| | +Del-zscore | $36.89_{\pm0.51}$ $(-6.87)$ | $14.41_{\pm0.36}$ $(-5.99)$ | $28.76_{\pm0.45}$ $(-5.89)$ |
| | +DefinitionEMB | $\mathbf{43.96}^{\ddagger}_{\pm0.43}$ $(+0.20)$ | $\mathbf{20.61}^{\ddagger}_{\pm0.32}$ $(+0.21)$ | $\mathbf{34.87}^{\ddagger}_{\pm0.51}$ $(+0.22)$ |
| **BillSum** | BART | $\mathbf{51.02}_{\pm0.28}$ | $32.44_{\pm0.40}$ | $39.11_{\pm0.26}$ |
| | +DelDirection | $50.89_{\pm0.29}$ $(-0.13)$ | $32.22_{\pm0.35}$ $(-0.22)$ | $38.97_{\pm0.21}$ $(-0.14)$ |
| | +Del-zscore | $45.56_{\pm0.07}$ $(-5.46)$ | $23.44_{\pm0.13}$ $(-9.00)$ | $32.14_{\pm0.02}$ $(-6.97)$ |
| | +DefinitionEMB | $50.96_{\pm0.68}$ $(-0.06)$ | $\mathbf{32.64}^{\ddagger}_{\pm0.25}$ $(+0.20)$ | $\mathbf{39.28}_{\pm0.16}$ $(+0.17)$ |

Table 8: Performance of BART on text summarization datasets (mean$_{\pm\text{std}}$ across 3 runs). (+score) indicates the improvement compared to the backbone. ‡ indicates that the score is significantly superior to the backbone with a p-value $< 0.05$.

| Model | ROUGE-1 | ROUGE-2 | ROUGE-L |
|---|---|---|---|
| T5 | $41.96_{\pm0.53}$ | $19.40_{\pm0.37}$ | $34.96_{\pm0.57}$ |
| +DelDirection | $41.81_{\pm0.17}$ $(-0.15)$ | $19.27_{\pm0.09}$ $(-0.13)$ | $34.77_{\pm0.20}$ $(-0.19)$ |
| +Del-zscore | $42.42_{\pm0.28}$ $(+0.46)$ | $19.66_{\pm0.14}$ $(+0.26)$ | $35.44_{\pm0.26}$ $(+0.48)$ |
| +DefinitionEMB | $\mathbf{42.81}^{\ddagger}_{\pm0.13}$ $\mathbf{(+0.85)}$ | $\mathbf{19.86}^{\ddagger}_{\pm0.08}$ $\mathbf{(+0.46)}$ | $\mathbf{35.88}^{\ddagger}_{\pm0.21}$ $\mathbf{(+0.92)}$ |
| Gemma | $16.71_{\pm0.45}$ | $\mathbf{2.11}_{\pm0.05}$ | $15.56_{\pm0.34}$ |
| +DelDirection | $15.70_{\pm0.15}$ $(-1.01)$ | $1.83_{\pm0.05}$ $(-0.28)$ | $14.60_{\pm0.07}$ $(-0.96)$ |
| +Del-zscore | $13.30_{\pm3.85}$ $(-3.41)$ | $2.08_{\pm0.65}$ $(-0.03)$ | $12.14_{\pm3.56}$ $(-3.42)$ |
| +DefinitionEMB | $\mathbf{16.94}^{\ddagger}_{\pm0.53}$ $\mathbf{(+0.23)}$ | $2.09_{\pm0.09}$ $(-0.02)$ | $\mathbf{15.75}^{\ddagger}_{\pm0.49}$ $\mathbf{(+0.19)}$ |

Table 9: Performance of T5 and Gemma on CNNDM (mean$_{\pm\text{std}}$ across 3 runs). (+score) indicates the improvement compared to backbones.

### 6.2.2 Performance on General Contexts

Beyond rare tokens, we must ensure DefinitionEMB preserves performance on frequent ones. This section evaluates the compatibility and robustness of the reconstructed embeddings by measuring their impact on general downstream performance. These results serve as a sanity check—verifying that our method integrates seamlessly with PLMs—and also provide evidence of its practical applicability beyond rare-token-focused settings. To evaluate this, we used public abstractive summarization datasets, including CNNDM, XSum, BillSum, and Y-BIGPATENT. Table 8 reveals three key insights: (1) DelDirection and Del-zscore Show Limited Gains. While DelDirection improves BART's ROUGE-L by 0.21 on XSum, it degrades performance on BillSum and shows negligible differences on other datasets. Similarly, Del-zscore shows benefits on T5 but degrades performance on other backbones. (2) DefinitionEMB's Consistent Improvements. By improving *semantic relatedness* for rare tokens, DefinitionEMB improves BART's ROUGE-L scores across all datasets (+0.21 CNNDM, +0.21 Y-BIGPATENT, +0.22 XSum, +0.17 BillSum). (3) DefinitionEMB's Effectiveness Across Architectures. In Table 9, DefinitionEMB boosts T5 and Gemma's performance, whereas DelDirection

degrades performance. These results demonstrate that DefinitionEMB's reconstructed embeddings integrate seamlessly with the remaining parameters of PLMs without compromising overall functionality.

| Model | ROUGE-1 | ROUGE-2 | ROUGE-L |
|---|---|---|---|
| BART+DefinitionEMB | $\mathbf{36.11}^{\ddagger}_{\pm 0.24}$ | $\mathbf{15.75}^{\ddagger}_{\pm 0.39}$ | $\mathbf{33.44}^{\ddagger}_{\pm 0.13}$ |
| w/o mimicking | $31.97_{\pm 0.75}$ | $11.82_{\pm 0.56}$ | $29.20_{\pm 0.75}$ |
| w/o definition | $12.01_{\pm 0.84}$ | $0.99_{\pm 0.32}$ | $10.87_{\pm 0.90}$ |
| w/o DefinitionEMB | $34.99_{\pm 0.57}$ | $14.67_{\pm 0.44}$ | $32.19_{\pm 0.57}$ |

Table 10: Ablation results ($\text{mean}_{\pm \text{std}}$ across 3 runs) in generating summaries with rare tokens.

### 6.2.3 Ablation Study

DefinitionEMB consists of two core components: (1) the definition corpus, which introduces rich semantic information used to reconstruct the embeddings, and (2) a mimicking objective that anchors reconstructed embeddings to the pretrained geometry. To isolate the contribution of each component in DefinitionEMB, we conduct an ablation study by selectively disabling its two core mechanisms. Appendix C.3 describes the detailed setting and additional results. Table 10 reports ROUGE scores averaged over three random seeds. Removing the mimicking objective leads to consistent performance degradation, indicating that geometric anchoring is critical for stable semantic reconstruction. Removing definition data causes a substantial drop across all metrics, showing that improvements rely on explicit definitional semantics rather than pretrained signals alone. Disabling both components reverts performance to the backbone level. Overall, DefinitionEMB achieves the best performance, confirming both components are necessary.

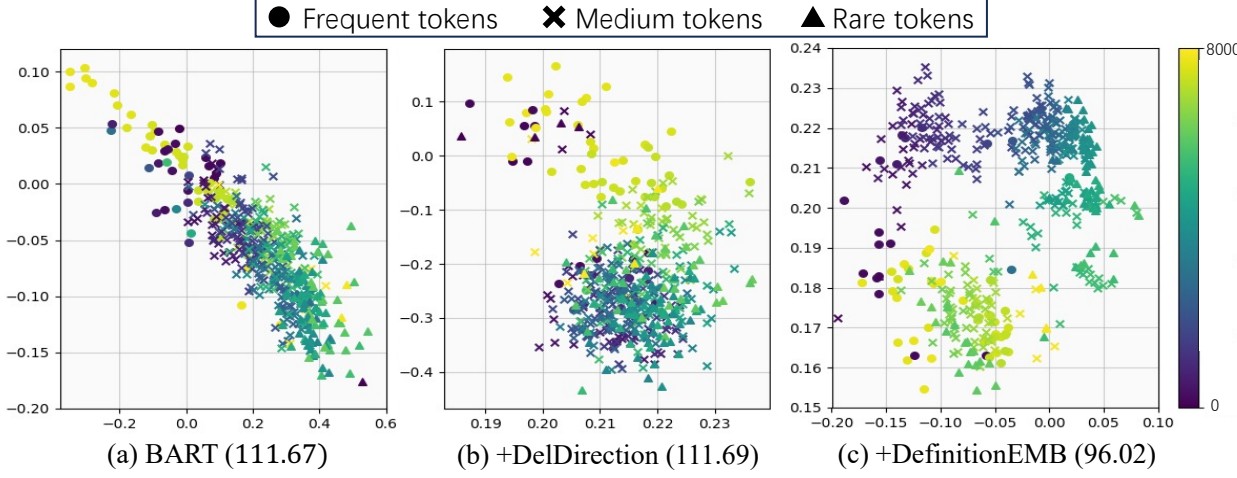

Figure 5: Projected embeddings of numeric tokens. ($\cdot$) shows the maximum distance between tokens.

## 6.3 Qualitative Analysis of Embedding Space

**Semantically Related Neighbors.** We evaluate whether tokens' neighbors are semantically related in the embedding space. Since PLM vocabularies primarily consist of subwords lacking explicit semantic definitions, direct evaluation is infeasible. Thus, we conducted an experiment on numeric tokens, whose semantic relationships (*i.e.,* numerical proximity) are quantifiable. Based on established findings that numeric features are continuously and monotonically represented (Heinzerling & Inui, 2024; Engels et al., 2025), we hypothesize that semantically correct representations of numbers should be closely mapped on the basis of their numeric value. Specifically, we extracted 492 numeric subwords from BART's vocabulary and visualized their embeddings via SVD. To quantify semantic relationships, we define a *Max Difference* metric: for each numeric token, we identify its three nearest numeric neighbors (in Euclidean space) and calculate the largest

numerical gap among them. DefinitionEMB achieves the lowest Max Difference (96.02), which reflects more coherent, definition-consistent numeric neighborhoods, rather than a collapse of numeric tokens into regions dominated by frequent tokens. Figure 5 visually confirms these results: numeric embeddings from both BART and DelDirection are scattered regardless of their numeric values. In contrast, DefinitionEMB produces a circular layout, increasing numerically in a clockwise direction. This reorganization suggests that token neighborhoods are formed based on definitional semantics rather than usage frequency, providing a deeper token-level understanding that directly supports the performance gains in downstream tasks. While this probe focuses on numeric subword tokens, our evidence is not restricted to numbers. Appendix C.1 together with a complementary case study in Figure 6 (a) further reveals that the rare token "Ġeverlasting" is surrounded by more *semantics-related* tokens than in both BART and DelDirection (Figure 2).

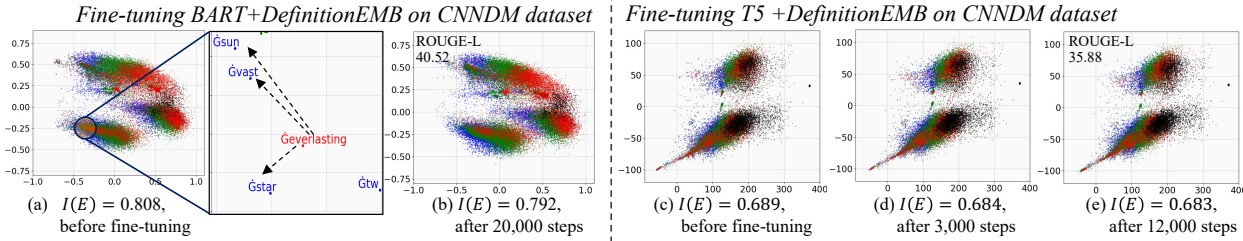

Figure 6: Projected token embeddings in BART+DefinitionEMB before and after fine-tuning. Embeddings with different frequencies maintain their distance during fine-tuning, demonstrating *geometry-preserving*.

**Geometry-Preserving.** Figure 6 depicts the fine-tuning dynamics of DefinitionEMB, where tokens exhibit minimal drift. This behavior aligns with PLMs in Figure 2, indicating that DefinitionEMB inherits the *geometry-preserving* property of PLMs. Additionally, after fine-tuning, the $I(\mathbf{E})$ values decrease only slightly (by $\leq 0.016$), indicating that isotropy is largely maintained throughout training. Collectively, the observations in Section 6.3 confirm that definitional semantics are successfully integrated into the well-pretrained geometry while inheriting its geometry-preserving property. Specifically, semantically related tokens form coherent clusters while preserving the overall distribution during fine-tuning.

| Source | ... **in the "Avengers: Age of Ultron" sequel**. But there was one character who remained a mystery: **the Vision, to be played by Paul Bettany**.... **Thursday was the eve of the new Netflix series "Daredevil," and after a photoshopped first look at Charlie Cox's iconic red Daredevil suit went out, Marvel put out a video of the real one.** Not to be outdone, **director Bryan Singer announced a new character for next year's sequel "X-Men: Apocalypse," by telling Empire magazine that Ben Hardy would be playing**... |
|---|---|
| Reference | Marvel Studios releases first looks at Paul Bettany as the Vision in "Avengers: Age of Ultron" and Charlie Cox in full "Daredevil" costume . Jamie Bell's character of The Thing was also unveiled for 20th ... the first look at "X-Men: Apocalypse" Angel played by Ben Hardy . |
| BART | Paul Bettany will play the Vision in the "Avengers: Age of Ultron" sequel . The actor has been playing the android for many years in the comics . The "Fantastic For" reboot's" The Thing" looks pretty much like The Thing we already knew . |
| +DelDirection | Paul Bettany's character in "Avengers: Age of Ultron" is finally revealed . The actor has been playing the Vision in the comics for many years . The "Fantastic Fou" reboot's" The Thing" looks pretty much like The Thing we already knew (but CGI) |
| +DefinitionEMB | Paul Bettany will play the Vision in the "Avengers: Age of Ultron" sequel . Marvel Studios also announced a new character for "X-Men: Apocalypse" Ben Hardy will play the winged mutant Angel in "X-Men: Apocalypse," director Bryan Singer said . |

Table 11: Sample summarization of CNNDM test set. **Bold** in source indicates the reference-related text. Underline in reference and model outputs indicates the rare token with index larger than 40,000 in PLM.

## 6.4 Case Study

Table 11 presents a case study showing how DefinitionEMB improves the generation of semantically important rare tokens. The source article and reference contain several rare entities like Daredevil and Apocalypse. Both BART and DelDirection fail to produce these terms and instead output frequent but less specific substitutes. DefinitionEMB successfully generates these rare entities while preserving factual correctness, demonstrating its improved ability to retrieve appropriate rare lexical items when required by context. The example provides qualitative evidence that DefinitionEMB enhances rare-token semantic accessibility rather than merely increasing their occurrence.

### 6.5 Future Work

While DefinitionEMB can be directly applied to token-level downstream tasks, a more comprehensive evaluation is needed to fully assess its benefits across a broader range of tasks. In particular, future work will explore tasks where rare-token representations play a critical role, such as POS tagging and machine translation. We also plan to explore how definition-based reconstruction interacts with task-specific fine-tuning dynamics and larger LLMs.

## 7 Conclusion

This work proposes DefinitionEMB, a novel and architecture-agnostic approach designed to address the semantic degradation of rare subword embeddings in Pretrained Language Models (PLMs). Our methodology reconstructs token embeddings using explicit dictionary definitions to inject reliable lexical semantics, and employs a mimicking objective to align these new representations with the PLM's native semantic space. This geometric regularization ensures that the reconstructed embeddings inherit the pretrained geometry-preserving property—our observation that PLM embeddings maintain their global structure during fine-tuning. By anchoring definition-derived semantics to this stable geometry, we effectively prevent the potential degeneration during subsequent fine-tuning. Experiments on RoBERTa, BART, T5, and Gemma show that DefinitionEMB consistently improves downstream performance. Our ablation study further confirms that the mimicking objective is essential. This demonstrates a key insight: aligning new semantic information with the PLM's native geometry is an effective approach to achieving robust semantic representation during fine-tuning.

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

# A Implementation Details

## A.1 Used Artifacts

All models and experiments are implemented using Fairseq (Ott et al., 2019) and HuggingFace Transformers (Wolf et al., 2020). Frameworks are used consistently across pretraining, DefinitionEMB reconstruction, and downstream fine-tuning to ensure reproducibility and fair comparison.

## A.2 Training DefinitionEMB

Table 12 summarizes the hyperparameters used to train DefinitionEMB initialized from RoBERTa, BART, T5 and Gemma. Unless otherwise stated, we follow the original optimization settings recommended by each model to avoid introducing confounders. All experiments are conducted with three independent random seeds and the reported results are averaged accordingly. For RoBERTa and BART, we adopt the Adam optimizer with inverse square root learning rate scheduling, consistent with their original pretraining configurations. For T5 and Gemma, we use Adafactor following standard practice. Dropout rates follow the backbone defaults. To reduce computational overhead, we freeze the embedding layer during DefinitionEMB training for BART, T5 and Gemma.

| Hyperparameters | RoBERTa | BART | T5 | Gemma |
|---|---|---|---|---|
| # of updates | 400,000 | 250,000 | 125,000 | 125,000 |
| # of warm-up updates | 24,000 | 20,000 | 0 | 0 |
| Batch size (tokens) | 4,096 | 4,096 | 8,192 | 8,192 |
| Learning rate | 1e-4 | 1e-4 | 1e-3 | 1e-3 |
| Warm-up init lr | 1e-7 | 1e-7 | N/A | N/A |
| Optimizer | Adam | Adam | Adafactor | Adafactor |
| Adam betas | (0.9, 0.98) | (0.9, 0.98) | N/A | N/A |
| Weight decay | 0.01 | 0.01 | N/A | N/A |
| Dropout rate | 0.1 | 0.1 | 0.1 | 0.0 |
| # of seeds | 3 | 3 | 3 | 3 |

Table 12: Hyperparameters for DefinitionEMB initialized from RoBERTa, BART, T5, and Gemma. N/A denotes models do not have this parameter.

| Hyperparameters | MRPC | RTE | QQP |
|---|---|---|---|
| # of updates | 2,296 | 2,036 | 113,272 |
| # of warm-up updates | 137 | 122 | 28,318 |
| Batch size (sentences) | 16 | 16 | 32 |
| Learning rate | 1e-5 | 2e-5 | 1e-5 |
| Optimizer | Adam | Adam | Adam |
| Adam betas | (0.9,0.98) | (0.9,0.98) | (0.9,0.98) |
| Weight decay | 0.1 | 0.1 | 0.1 |
| Dropout rate | 0.1 | 0.1 | 0.1 |
| # of seeds | 3 | 3 | 3 |
| Evaluation metric | Accuracy | Accuracy | N/A |

Table 13: Hyperparameters used for fine-tuning RoBERTa-related models across different datasets. N/A denotes models do not have this parameter.

## A.3 Fine-tuning on Downstream Tasks

Tables 13, 14 and 15 report the hyperparameters to fine-tune for the natural language understanding and summarization tasks. In Table 13, for GLUE-style classification tasks (MRPC, RTE, QQP), we follow the official train/validation/test splits and adopt the standard fine-tuning protocol used in prior RoBERTa studies. Optimization uses Adam, and performance is evaluated using accuracy on official test sets. All results are averaged over three random seeds.

| Hyperparameters | CNNDM (T5) | CNNDM (Gemma) |
|---|---|---|
| # of updates | 12,000 | 12,000 |
| # of warm-up updates | 0 | 0 |
| Batch size (tokens) | 65,536 | 65,536 |
| Learning rate | 1e-3 | 1e-3 |
| Optimizer | Adafactor | Adafactor |
| Dropout rate | 0.1 | 0.0 |
| # of seeds | 3 | 3 |
| Beam size | 4 | 4 |
| Length Penalty | 2.0 | 2.0 |
| Evaluation metric | Rouge-F1 | Rouge-F1 |

Table 14: Hyperparameters for T5 and Gemma.

In Table 14, for summarization on CNNDM, we fine-tune T5 and Gemma using the official dataset split. We use Adafactor optimization with a large token-level batch size and report ROUGE-F1 scores. During decoding, beam search is applied with a beam size of 4. All BART, T5 and Gemma employ weight tying during text generation, where the input embedding matrix $\mathbf{E}$ is reused as the output projection for computing logits. This design choice ensures consistency between reconstructed embeddings and generation behavior.

| Hyperparameters | MRPC | RTE | CNNDM | Y-BIGPATENT | XSUM | Billsum |
|---|---|---|---|---|---|---|
| # of updates | 700 | 1020 | 20,000 | 92,880 | 15,000 | 21,320 |
| # of warm-up updates | 42 | 61 | 500 | 7,430 | 500 | 1,705 |
| Batch size (sentences) | 64 | 32 | N/A | N/A | N/A | N/A |
| Batch size (tokens) | N/A | N/A | 65,536 | 8,192 | 32,768 | 8,192 |
| Learning rate | 2e-5 | 1e-5 | 3e-05 | 3e-5 | 3e-05 | 3e-5 |
| Optimizer | Adam | Adam | Adam | Adam | Adam | Adam |
| Adam betas | (0.9, 0.98) | (0.9, 0.98) | (0.9, 0.999) | (0.9, 0.999) | (0.9, 0.999) | (0.9, 0.999) |
| Weight decay | 0.01 | 0.01 | 0.01 | 0.01 | 0.01 | 0.01 |
| Dropout rate | 0.1 | 0.1 | 0.1 | 0.1 | 0.1 | 0.1 |
| # of seeds | 3 | 3 | 3 | 3 | 3 | 3 |
| Beam size | N/A | N/A | 4 | 4 | 6 | 4 |
| Length Penalty | N/A | N/A | 2.0 | 2.0 | 1.0 | 0.6 |
| Evaluation metric | Accuracy | Accuracy | Rouge-F1 | Rouge-F1 | Rouge-F1 | Rouge-F1 |

Table 15: Hyperparameters used for fine-tuning BART-related models across different datasets. N/A denotes models do not have this parameter.

Table 15 summarizes the fine-tuning configurations used for BART-related models across all evaluated datasets, including both classification and summarization tasks. We follow the official train/validation/test splits and adopt the standard fine-tuning protocol used in prior BART studies. For GLUE-style classification benchmarks (MRPC and RTE), models are fine-tuned using sentence-level batches and evaluated with accuracy. For summarization (CNNDM, Y-BIGPATENT, XSUM, and Billsum), we adopt token-level batching and evaluate performance using ROUGE-F1, following standard practice in abstractive summarization. For generation tasks, beam search decoding is employed with dataset-specific beam sizes and length penalties, consistent

with prior summarization studies. Experiments are conducted with three independent random seeds to ensure reliability of reported results.

Overall, these settings provide a complete and transparent specification of our experimental protocol. By using official splits, standard optimization practices, and multi-seed evaluation, we ensure that observed performance differences are attributable to DefinitionEMB rather than confounding factors in experimental design. This protocol aligns with standard PLM evaluation practice and supports fair and reproducible comparison across datasets.

### A.4 Experimental Settings in Figure 1

For the initialization experiment shown in Figure 1, which illustrates token embedding degeneration from scratch, we train BART-large using a learning rate of 0.001, a token batch size of 64,000, 50,000 total updates, and 4,000 warm-up steps. These settings follow standard large-scale language model training practices and are used solely for visualization purposes rather than downstream evaluation.

### A.5 Hyperparameter Setting of $\alpha$

| Baseline $\qquad\alpha$ | 1 | 5 | 10 | 20 |
|---|---|---|---|---|
| RoBERTa | 89.2 | **90.2** | 89.5 | 89.7 |
| BART | 88.2 | **90.2** | 88.7 | 89.2 |

Table 16: Accuracy for Baseline+DefinitionEMB with various $\alpha$ on the MRPC validation set.

| Baseline $\qquad\alpha$ | 3 | 5 | 10 | 20 |
|---|---|---|---|---|
| RoBERTa | 77.3 | **79.1** | 76.2 | 76.5 |
| BART | **85.9** | **85.9** | 84.8 | 84.1 |

Table 17: Accuracy for Baseline+DefinitionEMB with various $\alpha$ on the RTE validation set.

| $\alpha$ | ROUGE (F1) | | |
|---|---|---|---|
| | ROUGE-1 | ROUGE-2 | ROUGE-L |
| 10 | 44.45 | **21.62** | 41.17 |
| 30 | 44.30 | 21.44 | 41.04 |
| 50 | 44.23 | 21.37 | 40.99 |
| 100 | **44.62** | 21.54 | **41.40** |

Table 18: ROUGE for BART+DefinitionEMB with various $\alpha$ on the CNNDM validation set.

| $\alpha$ | ROUGE (F1) | | |
|---|---|---|---|
| | ROUGE-1 | ROUGE-2 | ROUGE-L |
| 5 | 44.02 | 20.66 | 34.95 |
| 10 | **44.22** | **20.95** | **35.24** |
| 20 | 43.81 | 20.55 | 34.78 |
| 100 | 42.46 | 19.18 | 33.62 |

Table 19: ROUGE for BART+DefinitionEMB with various $\alpha$ on the XSUM validation set.

| $\alpha$ | ROUGE (F1) | | |
|---|---|---|---|
| | ROUGE-1 | ROUGE-2 | ROUGE-L |
| 5 | 50.63 | 32.19 | 38.81 |
| 7 | 50.85 | **32.44** | **39.10** |
| 10 | **51.08** | 32.17 | 38.97 |
| 100 | 50.04 | 31.43 | 38.27 |

Table 20: ROUGE for BART+DefinitionEMB with various $\alpha$ on the Billsum validation set.

| $\alpha$ | ROUGE (F1) | | |
|---|---|---|---|
| | ROUGE-1 | ROUGE-2 | ROUGE-L |
| 10 | 43.62 | 18.53 | 37.43 |
| 20 | 43.93 | 18.84 | 37.74 |
| 30 | **44.22** | **19.12** | **38.03** |
| 100 | 42.96 | 17.76 | 36.73 |

Table 21: ROUGE for BART+DefinitionEMB with various $\alpha$ on the Y-BIGPATENT validation set.

The set of tokens chosen for replacement is determined by their frequency ranking in the original pretrained vocabulary $\mathcal{V}$. Our procedure is as:

1. DefinitionEMB Training: We first utilize a definition corpus to train the DefinitionEMB model.
2. Embedding Prediction: We use the trained DefinitionEMB to predict embeddings for all tokens in the vocabulary $\mathcal{V}$.
3. Selection and Replacement: The hyperparameter $\alpha$ defines the proportion of tokens selected for replacement. We select the bottom $\alpha$ percent of the vocabulary (i.e., the rarest tokens) and replace their original embeddings in the backbone PLM (e.g., BART) with the reconstructed embeddings from DefinitionEMB.
4. Fine-tuning: We fine-tune this modified backbone on the downstream task (e.g., CNNDM) and evaluate the model performance.

The optimal $\alpha$ is decided based on the fine-tuning performance on the corresponding validation set for each downstream task. Tables 16 to 21 display our tuning results. We observe that the optimal value of $\alpha$ is intrinsically linked to data density of the downstream task. A high $\alpha$ implies an aggressive replacement (e.g., $\alpha = 100$) of a large portion of the vocabulary's parameters. To ensure these new parameters integrate and coordinate with the existing global parameters, sufficient downstream data is required for effective optimization.

### A.6 Prompt Settings for DefinitionEMB

In Figure 7(a), to include a complete definition for better re-construction, our full prompt incorporates the definition, part-of-speech, capitalization, and case sensitivity of word $\boldsymbol{w}$, along with the tokenizer specific settings.

## B Datasets

### B.1 Numbers and Named Entities Definitions

We add definitions for 1,252 / 227 numbers in BART / T5 vocabulary $\mathcal{V}$ by translating numbers into their corresponding words, such as "2" to "two". Furthermore, we add definitions for 136 / 88 named entity tokens in BART / T5 vocabulary $\mathcal{V}$, such as "ĠNVIDIA", based on their Wikipedia pages or Google search results. These definitions are available at our anonymous Github `https://anonymous.4open.science/r/DefinitionEMB-93B7`.

## (a) Full prompt

> The definition of [word] is [definition] . Its part-of-speech ,
> bpe-form without space , capitalization , and uppercase are [pos] ,
> [wospace-word] , [cap-word] , and [upper-word] , respectively .

## (b) Corrupted prompt

> The definition of {corruption} is [definition] . Its part-of-speech ,
> bpe-form without space , capitalization , and uppercase are [pos] ,
> {corruption} , {corruption} , and {corruption} , respectively .

Figure 7: Constructed prompts. Brackets [] are a placeholder for the given word and its information. Texts with the same color indicate positions of a prompt and corresponding word information. {corruption} indicates the span for corrupted tokens. The bpe-form without space refers to the word's surface-form after removing the symbol that indicates the subword is at the beginning of a word.

## B.2 Rare Token Subset

To construct the rare token subset that is used in Table 6, we first filter the CNNDM test set to include target sentences whose tokens all appear in the training set. Additionally, for BART-related models, each filtered target sentence must contain at least 5% rare tokens with indices larger than 40,000 in $\mathcal{V}$, and these tokens' embeddings can be replaced by DefinitionEMB. For T5-related models, each filtered target sentence must contain at least 8.3% rare tokens with indices larger than 24,000 in $\mathcal{V}$. For Gemma-related models, each filtered target sentence must contain at least 31% rare tokens with indices larger than 200,000 in $\mathcal{V}$. This process finally yields 65 / 60 / 102 pairs of data for BART / T5 / Gemma-related models. This subset is also available at our anonymous GitHub.

## B.3 Probing Numeric-Related Semantics.

| Label | # of training | # of validation | # of test |
|-------|---------------|-----------------|-----------|
| Numeric <1000 | 836 | 93 | 72 |
| Numeric >1000 | 878 | 98 | 6 |
| Others | 900 | 100 | 78 |
| All | 2614 | 291 | 156 |

Table 22: Numeric dataset statistics.

Table 22 shows dataset statistics for this probing experiment. We first introduce our test set. As mentioned in Appendix B.1, definitions for 1,252 numbers in $\mathcal{V}$ are generated by translating numbers into their corresponding words. Out of these 1,252 numbers, 78 numeric tokens start with "Ġ" and appear in the CNNDM dataset with an index greater than 40,000 in the vocabulary $\mathcal{V}$. We also randomly select 78 non-numeric tokens from $\mathcal{V}$, ensuring they start with "Ġ" and belong to the rare group within the CNNDM dataset. These 156 tokens make up our test set. For the training and validation sets, we first randomly select 1,000 non-numeric tokens from the Wiktionary dataset. Similarly, we select numbers in the range [0, 1999] as our numeric words. These words were then divided into training and validation sets in a 9:1 ratio. We further filter the data to ensure no overlap between the training/validation and the test data.

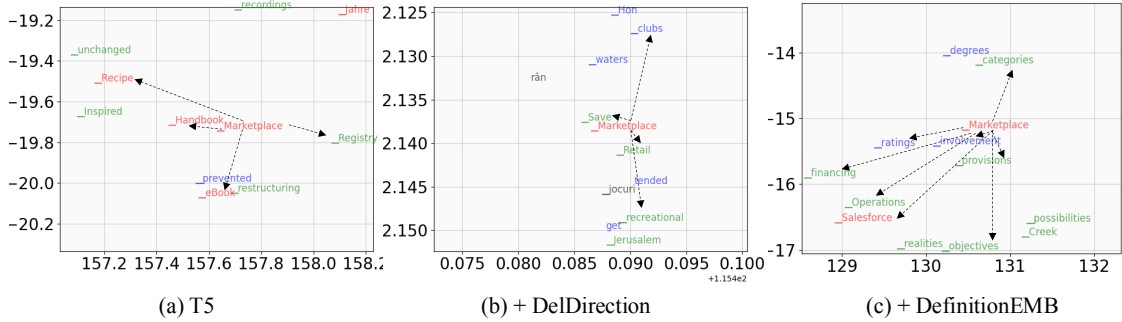

(a) T5       (b) + DelDirection       (c) + DefinitionEMB

Figure 8: Case study of the token embeddings of the token "_Marketplace" and its surrounding tokens. The dashed lines from "_Marketplace" point to its semantics-related tokens. The prefix "_" indicates that a subword is at the beginning of a word in the T5 BPE tokenizer.

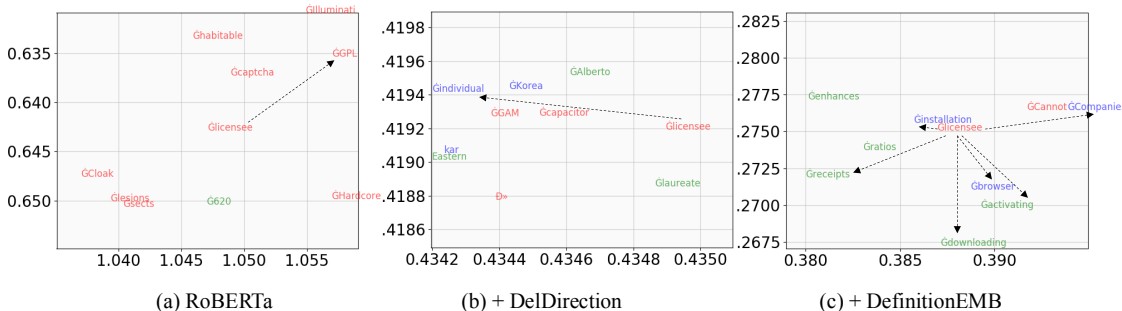

(a) RoBERTa       (b) + DelDirection       (c) + DefinitionEMB

Figure 9: Case study of the token embeddings of the token "Ġlicensee" and its surrounding tokens. The prefix Ġ indicates that a subword is at the beginning of a word in the BART/RoBERTa BPE tokenizer.

## C Complementary Results

### C.1 Geometric Distribution of Embeddings

**PLMs Lacks of Semantics.** Figures 8 and 9 show that the embeddings of both T5 and RoBERTa lack semantics for low-frequency tokens. By applying our proposed DefinitionEMB, more semantically related tokens are mapped into the same region.

**Geometry-Preserving of RoBERTa.** We trained RoBERTa-based models on the Quora Question Pairs (QQP) dataset Sharma et al. (2019b) for 113,272 steps. We replaced 3% of the last tokens from the vocabulay when using DefinitionEMB. Figure 10 shows that applying DelDirection to RoBERTa causes the spread of popular tokens from the original center. However, both RoBERTa and DefinitionEMB's embeddings exhibit geometry-preserving during fine-tuning. Although RoBERTa, T5, and BART have different model architectures (encoder-only vs. encoder-decoder), scales, and pre-training strategies, all they show similar geometry-preserving against representation degeneration.

### C.2 Isotropic Distribution

We measured the isotropy of the original PLMs' embeddings **E** and after completely replacing **E**. Table 23 shows that all PLMs exhibit high isotropy in the frequent group, but much lower isotropy in the rare group, indicating a constrained representational space for rare tokens. DelDirection improves overall isotropy and results in the highest isotropy for the rare group. Although it enables better spatial separation of rare tokens, the resulting embeddings lack semantic relatedness, as we further demonstrate in Section 6. This serves as evidence to show that methods such as DelDirection, which only address anisotropy, do not yield semantic benefits on their own. This observation challenges the prevailing assumption that anisotropy is the primary

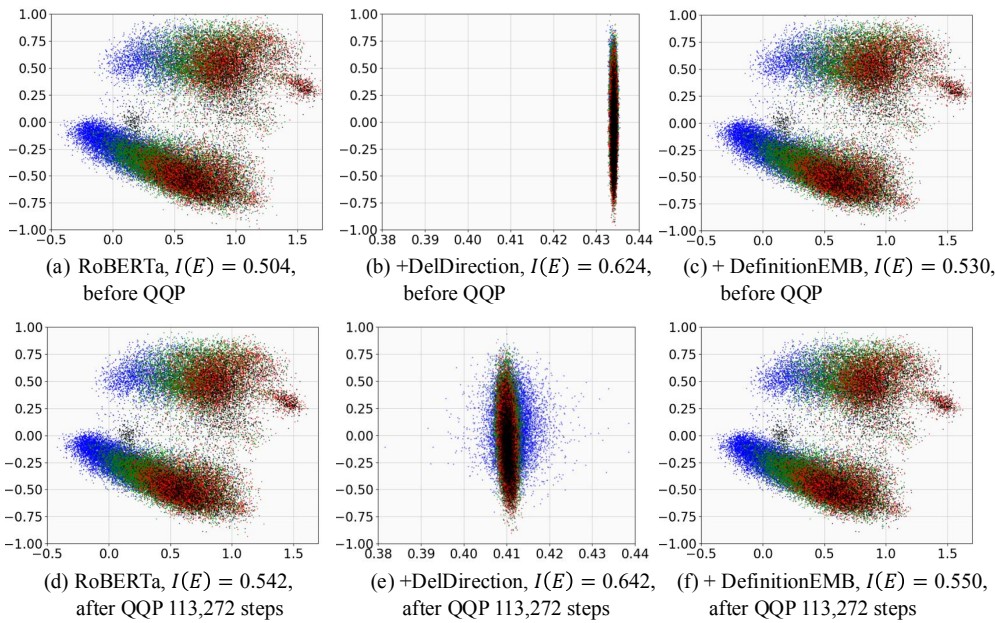

Figure 10: Projected token embeddings of RoBERTa-related models "before" and "after" fine-tuning on the QQP datasets.

cause of embedding degeneration and serves as the core motivation for DefinitionEMB—a method focused on injecting explicit semantics while preserving the embedding manifold, rather than merely reconfiguring the geometric distribution. DefinitionEMB also improves the isotropy, particularly for rare tokens, suggesting that it utilizes the embedding space more effectively than PLMs. Unlike DelDirection, DefinitionEMB aligns the embeddings with semantic information, as demonstrated in Section 6.

| Model | Frequent | Medium | Rare | All Tokens |
|---|---|---|---|---|
| RoBERTa | **0.694** | 0.501 | 0.315 | 0.504 |
| + DelDirection | 0.639 | **0.641** | **0.599** | **0.624** |
| + Del-zscore | 0.020 | 0.083 | 0.055 | 0.106 |
| + DefinitionEMB | $0.649_{\pm0.038}$ | $0.470_{\pm0.022}$ | $0.382_{\pm0.029}$ | $0.519_{\pm0.027}$ |
| BART | **0.851** | 0.668 | 0.515 | 0.751 |
| + DelDirection | 0.790 | 0.775 | **0.731** | 0.788 |
| + Del-zscore | 0.014 | 0.007 | 0.000 | 0.036 |
| + DefinitionEMB | $0.834_{\pm0.004}$ | $\mathbf{0.800}_{\pm0.025}$ | $0.603_{\pm0.006}$ | $\mathbf{0.876}_{\pm0.037}$ |
| T5 | 0.730 | 0.682 | 0.635 | 0.670 |
| + DelDirection | **0.747** | **0.746** | **0.746** | **0.747** |
| + Del-zscore | 0.998 | 1.000 | 0.997 | 1.000 |
| + DefinitionEMB | $\mathbf{0.747}_{\pm0.005}$ | $0.711_{\pm0.013}$ | $0.675_{\pm0.016}$ | $0.689_{\pm0.006}$ |
| Gemma | **0.553** | 0.385 | 0.361 | **0.489** |
| + DelDirection | 0.487 | **0.487** | **0.487** | 0.487 |
| + Del-zscore | 0.001 | 0.025 | 0.015 | 0.023 |
| + DefinitionEMB | $0.541_{\pm0.069}$ | $0.418_{\pm0.015}$ | $0.392_{\pm0.08}$ | $0.484_{\pm0.033}$ |

Table 23: Isotropy score of token embedding **E**. The frequent (30%), medium (50%), and rare (20%) groups are classified based on the token index in $\mathcal{V}$.

Importantly, DefinitionEMB does not aim to uniformly maximize isotropy across the entire embedding space. Instead, it selectively improves geometric structure only where pretrained rare-token embeddings are degraded,

ensuring that lexical distinctions in well-trained regions are preserved. As a result, performance and isotropy gains may vary across PLMs depending on how well their original embeddings represent rare tokens (e.g., BART vs. RoBERTa), while still yielding consistent improvements in downstream tasks involving such tokens.

### C.3 Ablation Study

We conduct three ablation studies to assess the contribution of each component of DefinitionEMB:

- Ablating Mimicking: To isolate the effect of the mimicking objective, we fine-tune the BART model on the same definition corpus used in DefinitionEMB, but using the standard next-token prediction objective. Since most PLM embeddings are originally trained via this objective, it serves as the most direct baseline to demonstrate that mere exposure to definitions via standard training is insufficient to recover rare-token semantics.

- Ablating Definition: To isolate the impact of definitional semantics, we substitute the definition corpus with examples extracted from the Wiktionary dataset, while retaining the mimicking objective. This transforms the task into context-based representation learning. The aim is to test the efficacy when learning token representations solely from minimal context (each rare token appears in only a single unique example), thereby underscoring the necessity of utilizing a definition-based corpus. This yielded a final dataset of 48,959 examples for BART.

- Ablating DefinitionEMB: This setting reverts the model to the original BART without any DefinitionEMB components.

| Model | ROUGE-1 | ROUGE-2 | ROUGE-L |
|---|---|---|---|
| BART+DefinitionEMB | $\mathbf{43.78}^{\ddagger}_{\pm 0.09}$ | $\mathbf{20.94}_{\pm 0.09}$ | $\mathbf{40.52}^{\ddagger}_{\pm 0.09}$ |
| w/o mimicking | $40.38_{\pm 0.04}$ | $17.77_{\pm 0.05}$ | $37.08_{\pm 0.04}$ |
| w/o definition | $17.95_{\pm 1.38}$ | $2.55_{\pm 0.49}$ | $16.21_{\pm 1.25}$ |
| w/o DefinitionEMB | $43.57_{\pm 0.27}$ | $20.93_{\pm 0.20}$ | $40.31_{\pm 0.26}$ |

Table 24: Ablation results (mean$_{\pm \text{std}}$ across 3 runs) in generating CNNDM summaries.

| Model | Frequent | Medium | Rare | All Tokens |
|---|---|---|---|---|
| BART+DefinitionEMB | 0.834 | **0.800** | 0.603 | **0.876** |
| w/o mimicking | 0.474 | 0.464 | 0.443 | 0.450 |
| w/o definition | 0.751 | 0.770 | **0.614** | 0.711 |
| w/o DefinitionEMB | **0.851** | 0.668 | 0.515 | 0.751 |

Table 25: Comparison of Isotropy scores of **E**.

We then evaluate the resulting model on the same downstream tasks as in our main experiments. All hyperparameters and optimization settings are aligned with our standard setup to ensure a fair comparison. The results in Table 24 and Table 25 show consistent performance degradation after ablation, indicating that removing either core component of DefinitionEMB weakens its ability to improve isotropic distribution and semantic relatedness. Specifically, ablating the mimicking objective causes a significant isotropy decline across all frequency tiers, while retaining it substantially improves local isotropy for medium and rare tokens, showing that this objective is critical for aligning rare tokens with the pretrained geometry. Ablating the definitions leads to a smaller isotropy drop, with the score for all tokens decreasing from 0.876 to 0.751, suggesting that example contexts can still provide useful geometric anchors when combined with the mimicking objective. However, performance on CNNDM drops sharply to 16.21 ROUGE-L, indicating that example contexts alone capture limited semantics and further highlighting the advantage of dictionary definitions for rare tokens.

