# OpenReview forum: "Reconsidering Degeneration of Token Embeddings with Definitions"
_TMLR — Decision pending for TMLR_

### Review · Reviewer_4YvK · 2026-04-24

**Summary Of Contributions:**

The paper studies the fine-tuning of the token embeddings learned by pretrained
language model (PLM). The authors claim that current approaches to training and
fine-tuning language models result in an anisotropic embedding where rare
(sub)tokens are not embedded near semantic neighbors. They introduce a method to
learn semantic neighbors of rare tokens called DefinitionEMB that improves the
performance of PLMs in tasks pertaining to rare words/tokens.

Their main three contributions are:

- they provide an analysis into the cause of anisotropy by fine-tuning of PLMs.
- they introduce a method to fine-tune PLMs that preserve semantic
  meaningfulness between rare tokens.
- they evaluate their methods experimentally against other baseline approaches
  to provide evidence of its improvements.

**Audience:**

No

**Audience Explanation:**

I do not believe the findings of this paper would be of interest to the TMLR
audience. My previous comments elaborate why I do not think the results are
impactful.

**Broader Impact Concerns:**

No broader impact concerns.

**Claims And Evidence:**

No

**Claims Explanation:**

- One of the main claims made in the paper is that preserving geometry is more
  important than maintaining isotropy in achieving robust semantic
  representation (this is even the last line of the paper). However, there
  actually is not anything in the paper that supports this claim. While uSection
  4 discusses this briefly, there are no experiments that significantly support
  this statement. If the authors state that the results of their approach does
  so, I cannot agree with that (more on this next).

- It is hard to argue the fairness of the empirical evaluations used to
  demonstrate the strength of their method. I may have misunderstood, but their
  method just seems to be adding contexts for tokens that have fewer contexts in
  the corpus used to train a PLM. It is difficult to argue whether the
  improvement arises from anything their approach is doing other than adding more
  training data.

- I also cannot agree that the results actually show significance in the
  improvement made by DefinitionEMB. There are a few gripes I have about the
  experiments:
  - the standard deviation is not a valid measure for significance.
  - significance testing over only 3 runs is uninformative.
  - if you can only perform three runs, you might as well show the results of
    all 3 runs.

- Again for the later experiments, it is hard for me to agree that the results
  do indeed show that semantic meaningfulness is obtained by anything other
  having more training data on the rare tokens.

**Requested Changes:**

- Firstly, you have to have more than 3 runs. You cannot make any statistically
  significant statements using only 3 runs.

- You need to design experiments to show that it is not just adding more data is
  responsible for the increased performance.

---

> ### Author Response · Authors · 2026-05-09
> **Part I:  Reply to Reviewer 4YvK**
>
> ***We sincerely thank you for taking the time to carefully read our manuscript and for providing helpful comments. We address each of your concerns point-by-point below and have revised the paper accordingly to further improve clarity and overall quality.***
>
> **Q-1**: One of the main claims is that preserving geometry is more important than maintaining isotropy in achieving robust semantic representation. However, there is not anything in the paper that supports this claim...
>
> **Reply-1**: We thank the reviewer for this comment. We acknowledge that the original phrasing of this claim was ambiguous and did not clearly point to the supporting evidence. We address both issues below.
> - **Meaning of our claim**. Our claim concerns the role of the mimicking objective, which aligns newly injected definition-based representations with the PLM's native embedding space (i.e., newly injected representations are clustered with semantically related ones in embedding geometry while inheriting the geometry-preserving.)
> The point we wish to convey is that aligning geometry is an effective approach to achieving robust semantic representation during fine-tuning.
> - **Supporting evidence in our experiments**. Our ablation in Section 6.2.3 (Table 9) directly addresses this. We compare DefinitionEMB with a variant that uses the same definition data but without the mimicking objective. Specifically, by fine-tuning BART on the same definition corpus using the standard next-token prediction objective.
> The variant without mimicking yields a 4.24-point drop in ROUGE-L on BART, providing direct evidence that using mimicking to align geometry, not the use of definition data alone, is essential for the observed improvements.
> - **Refined phrasing**. To make our claim more precise and aligned with the evidence, we have revised the concluding statement from
>     > "This demonstrates a key insight: maintaining the embedding geometry is a more effective approach to achieving robust semantic representation during fine-tuning than purely addressing isotropy"
>
>     to:
>
>     > "Our ablation study further confirms that the mimicking objective is essential. This demonstrates a key insight: aligning new semantic information with the PLM's native geometry is an effective approach to achieving robust semantic representation during fine-tuning."
>
>
> **Q-2**: It is hard to argue the fairness of the empirical evaluations used to demonstrate the strength of their method. I may have misunderstood, but their method just seems to be adding contexts for tokens that have fewer contexts in the corpus used to train a PLM. It is difficult to argue whether the improvement arises from anything their approach is doing other than adding more training data.
>
> **Reply-2**: We thank the reviewer for this concern, which our ablation studies (Sec 6.2.3, Table 9) are specifically designed to address. We have ablated two key components of our method to demonstrate the effectiveness of each:
>
> - **Ablation 1: replacing the mimicking objective with standard next-token prediction.** This variant uses the same definition data as DefinitionEMB but applies it through standard language modeling fine-tuning. In effect, this is a direct test of "what if we just train on definitions as additional data?" The variant performs 4.24 ROUGE-L points worse than DefinitionEMB on BART, demonstrating that the gain does not come from the definition data itself, but from the mimicking objective that integrates the data.
>
> - **Ablation 2: replacing definitions with corpus-style examples.** This variant keeps the mimicking objective but replaces dictionary definitions with example sentences extracted from Wiktionary. These examples consist of naturally-occurring sentences containing the target token, designed to simulate the context-based learning a PLM receives during pretraining. We provide one example sentence per rare token. The variant performs 24.31 ROUGE-L points worse than DefinitionEMB on BART. This demonstrates that the type of data matters.
>
> - **A note on corpus-style examples.** We acknowledge that DefinitionEMB does use more total textual data than Ablation 2. In our setup, multiple definitions per token versus a single example per token. This setting, however, reflects a fundamental property of the rare-token that motivates our work: dictionary definitions are systematically available for rare tokens (since lexicographic resources cover them by design), while naturally-occurring corpus examples are scarce precisely because the tokens are rare. Using a single example per token in Ablation 2 is meant to reflect this practical scarcity, not to construct an artificially impoverished baseline. The reviewer's observation that more data is involved is therefore correct, but it does not undermine our claim: the contribution of our method is to demonstrate that dictionary definitions, a more available resource for rare tokens, can be effectively leveraged through a mimicking objective.

---

> ### Author Response · Authors · 2026-05-09
> **Part II: Reply to Reviewer 4YvK**
>
> **Q-3**: I also cannot agree that the results actually show significance in the improvement made by DefinitionEMB. There are a few gripes I have about the experiments: (i) the standard deviation is not a valid measure for significance; (ii) significance testing over only 3 runs is uninformative; (iii) if you can only perform three runs, you might as well show the results of all 3 runs. Again for the later experiments, it is hard for me to agree that the results do indeed show that semantic meaningfulness is obtained by anything other having more training data on the rare tokens.
>
> **Reply-3**: We thank the reviewer for raising these concerns about statistical methodology. We believe these points reflect an unclear description in our paper and we address each below.
>
> - **On standard deviation as a significance measure.** The reviewer is correct that standard deviation is a dispersion measure, not a significance test. In our paper, standard deviation is reported only as a descriptive statistic of across-seed performance variation; **statistical significance is tested via paired bootstrap (Koehn, 2004) on test set instances, not via standard deviation.**
>
> - **Clarification of our significance test.** For text summarization task, we utilize the bootstrap (Koehn, 2004) to perform significance test. Specifically, **we resample the test set 1,000 iterations with replacement.** For each iteration, we calculate how often our proposed method outperforms the baseline to determine the p-value. Since we use three different random seeds, before resample, **we aggregate the results from all seeds to form a combined test pool.** This approach simultaneously accounts for variations in both the random seeds and the data samples.
>
> - **Modification in our paper.** To help audience better understand this significance test, we modify our draft as follows:
> “Statistical significance is computed using paired bootstrap resampling (Koehn, 2004). Specifically, we resample the test set 1,000 iterations with replacement. Before resample, we aggregate the results from all seeds to form a combined test pool. This approach simultaneously accounts for variations in both the random seeds and the data samples.”
>
> **Reference:**
>
> Koehn, Philipp. Statistical Significance Tests for Machine Translation Evaluation. EMNLP,  2004.

---

### Review · Reviewer_WJng · 2026-04-26

**Summary Of Contributions:**

LLM embeddings often exhibit anisotropy, and in particular, it has been observed that tokens with similar frequencies are close in embedding space, even though they are semantically unrelated. The authors conduct experiments to show that anisotropy is not the fundamental issue, but that rare tokens lack semantic meaning. The authors propose DefinitionEMB to inject semantic meaning from dictionary definitions. Then, the method is tested on a variety of architectures and datasets.

**Audience:**

Yes

**Audience Explanation:**

Yes, the paper seems to be novel, and address concerns that have not been studied in prior literature.

**Claims And Evidence:**

Yes

**Claims Explanation:**

Yes, all claims are backed by experimental evidence. The authors also provide sufficient details needed to reproduce the experiments.

**Requested Changes:**

The paper is generally well-written and easy to follow. Also, to disclose, my work is primarily not in LLMs.

- I don’t think the runtime of adding DefinitionEMB is reported for the experiments. This should be mentioned this as it seems the method may be more computationally expensive compared to the baselines.
- Cloze exercise should be defined? (Section 5.1)
- 3 independent seeds (trials) does not seem enough; I would recommend doing at least 10 (if compute is not an issue).

The first point will be important consideration for my recommendation.

---

> ### Author Response · Authors · 2026-05-10
> **Part-I: Reply to Reviewer WJng**
>
> ***Thank you very much for your valuable and constructive comments. We appreciate your careful summary of our work and your recognition of our motivation. We have carefully addressed your concerns and revised the manuscript accordingly, and we hope the revisions further clarify DefinitionEMB’s motivation and effectiveness.***
>
>
> **Question-1**: I don’t think the runtime of adding DefinitionEMB is reported for the experiments. This should be mentioned this as it seems the method may be more computationally expensive compared to the baselines.
>
> **Reply-1**: We thank the reviewer for suggesting that we report the runtime of DefinitionEMB. The table below shows the comparison using 4 NVIDIA A100 (80GB) GPUs. The downstream tasks are evaluated on the CNN/DailyMail (CNNDM) dataset.
>
> | Model | Pre-training DefinitionEMB | Downstream Fine-tuning | Downstream Inference (Per Sample) |
> |---|---:|---:|---:|
> | T5 | - | 12.1h | 5.2 sec |
> | +DelDirection | - | 11.9h | 5.2 sec |
> | +Del-zscore | - | 12.4h | 4.6 sec |
> | +DefinitionEMB | 12.0h | 12.2h | 4.6 sec |
> | BART | - | 9.6h | 3.6 sec |
> | +DelDirection | - | 10.8h | 4.0 sec |
> | +Del-zscore | - | 9.5h | 2.0 sec |
> | +DefinitionEMB | 8.4h | 9.8h | 3.6 sec |
> |
>
> We acknowledge that DefinitionEMB requires substantially pre-training time. However, we emphasize three properties that justify this cost:
> - **One-time cost, reused across all downstream tasks.** DefinitionEMB is trained once per PLM. Once trained, DefinitionEMB is applied to reconstruct embeddings via a single forward pass per rare token; the reconstructed embeddings are then substituted into the PLM's embedding matrix. After substitution, the modified PLMs can be applied across all downstream tasks.
> - **Fine-tuning and inference run at the same speed as a vanilla PLM.** After substitution, the modified PLM has identical architecture and dimensions to the original. Therefore, downstream fine-tuning and inference proceed at the same speed as a vanilla PLM, with no additional computational overhead.
>
> We have revised our paper and added runtime comparison in Section 6.1 as follows:
>
> - "Table 4 lists the runtime statistics for BART and T5, where model training are conducted on 4 NVIDIA A100 (80GB) GPUs. Although DefinitionEMB requires substantial pre-training time, this is a one-time cost. Once trained, DefinitionEMB can be directly applied during inference with high efficiency."
>
>
> **Question-2**: Cloze exercise should be defined? (Section 5.1)
>
> **Reply-2**: We thank the reviewer for this suggestion. We will update Section 5.1 with the following definition:
> - "A cloze exercise is a task where a word in a sentence is replaced with a placeholder (e.g., a \<MASK\> token), and the model is required to predict the replaced word from the surrounding context. Our training adopts this paradigm in the form of a denoising autoencoder (Vincent et al., 2010)."

---

> ### Author Response · Authors · 2026-05-10
> **Part-II: Reply to Reviewer WJng**
>
> **Question-3**: 3 independent seeds (trials) does not seem enough; I would recommend doing at least 10 (if compute is not an issue).
>
> **Reply-3**: We thank the reviewer for this suggestion. We believe this concern arises from an unclear description of our significance methodology, which we now clarify.
>
> - **Our significance test does not rely on the number of seeds.**
>     - First, for the text summarization task, we apply the paired bootstrap (Koehn, 2004) to perform significance testing. Specifically, we resample the test set 1,000 times with replacement, and for each resample, we compute the difference between new method and the baseline. Thus, the bootstrap distribution comprises 1,000 resampled differences, not three seed-level scores.
>     - Second, **the role of multiple seeds in our design is to capture training stochasticity** (data order, dropout), not to serve as the unit of distribution. We aggregate the outputs from all three seeds into a combined pool, and the bootstrap resamples from this pool.  This design simultaneously captures variation from both test sample composition and stochastic training runs. Increasing from 3 to 10 seeds would scale the pool linearly but would not qualitatively change the bootstrap power.
>
> - **Compute considerations.** Our experiments span 4 PLMs (RoBERTa, BART, T5, Gemma), each fine-tuned on multiple downstream datasets. The text summarization task involves long input sequences, making fine-tuning particularly resource-intensive: for instance, fine-tuning Gemma on the CNN/DailyMail dataset requires approximately 18 hours on 4 NVIDIA A100 (80GB) GPUs per seed. Across 4 baselines x 6 PLM-dataset combinations, running on additional 7 seeds would require approximately 126 days of additional compute on 4 NVIDIA A100 (80GB) GPUs. This is beyond our available compute budget. Given that the additional seeds would not strengthen the bootstrap inference, we believe the current configuration is statistically sufficient for the claims we make.
>
> We have revised Section 6.1 of the paper to describe the bootstrap methodology more precisely as follows:
>
> - "Specifically, we resample the test set 1,000 iterations with replacement. Before resample, we aggregate the results from all seeds to form a combined test pool. This approach simultaneously accounts for variations in both the random seeds and the data samples."

---

### Review · Reviewer_FzmM · 2026-04-27

**Summary Of Contributions:**

The paper is concerned with the embeddings of rare subword tokens. The major contribution of the paper is to propose a new method called DefinitionEMB which preserves semantically meaningful embeddings as well as geometry preservation. The paper carries out empirical validation to demonstrate its validity.

**Audience:**

Yes

**Audience Explanation:**

I still think the paper is worth reading. But this judgment is better made but someone who worked on embeddings and anisotropy.

**Broader Impact Concerns:**

No need to discuss ethical implications.

**Claims And Evidence:**

No

**Claims Explanation:**

I don't specialize in the area, but I found the paper not easy to follow. Here are my main points:


1-If the objective is a focus on **rare** subword tokens then does it really matter? (1) By definition they are rare. (2) Isn’t the semantic meaning of subwords especially more dependent on the context?

2-Why didn’t the authors motivate the objective behind their paper through a concrete clear example at the beginning?

3-I don't understand the first point in the contribution on the PLMs geometry-preservation and anisotropy. Isn't this already known?

4-Using methods like sentenceBert through contrastive learning, don't we end up with semantically meaningful embeddings?

**Requested Changes:**

Please see address the points I made above.

---

> ### Author Response · Authors · 2026-05-09
> **Part-I: Reply to Reviewer FzmM**
>
> ***Thank you very much for your valuable and constructive comments. We sincerely appreciate your efforts to help improve the quality of our work. We have carefully responded to your concerns and hope that our revisions will help alleviate them.***
>
> **Question-1**: If the objective is a focus on rare subword tokens then does it really matter? (1) By definition they are rare. (2) Isn’t the semantic meaning of subwords especially more dependent on the context?
>
> **Reply-1**: We thank the reviewer for this constructive feedback. We address the two points as follows:
> - **Rare subword tokens often carry critical semantics.** Although they appear infrequently, rare subwords are often the core of named entities, numbers, and technical terms. In fields like news or medicine, errors on these tokens can lead to serious problems. For example, consider a model summarizing the sentence: "102 people were injured in the accident." The subword token for "02" is rare, and its input embedding is clustered indistinguishably with other rare numerals, such as “0000”. As a result, the model may incorrectly output "10,000 people were injured". This turns a moderate incident into a catastrophe simply because the model could not tell the difference between two rare subwords.
>
> - **Contextual meaning depends on high-quality input embeddings.** We agree that a subword's full meaning depends on its full-word (context). However, we want to clarify how this process works in Transformer architecture. In this architecture, input tokens are first converted into input embeddings (which our work focuses on). These are then processed by attention layers to create contextualized embeddings. If the initial input embeddings for rare subwords lack clear semantic meaning, the resulting contextualized representations will also be flawed. Therefore, improving input embeddings is essential because they are the basic building blocks that the model uses to create context. Without good building blocks, the contextualization process cannot function correctly.
>
> **Question-2**: Why didn’t the authors motivate the objective behind their paper through a concrete clear example at the beginning?
>
> **Reply-2**: We thank the reviewer for this insightful suggestion. We agree that adding a concrete example will help readers better understand our motivation. In the revised version, we will incorporate the numeric-summarization example mentioned in our first response into the **Introduction**. The added text is as follows:
> - “For example, consider a model summarizing the sentence: ‘*102 people were injured in the accident.’* The subword token for ‘02’ is rare, and its input embedding is clustered indistinguishably with other rare numerals, such as ‘0000’. As a result, the model may incorrectly output ‘*10,000 people were injured’*. This turns a moderate incident into a catastrophe, simply because the model could not tell the difference between two rare subwords.”
>
> **Question:-3**: I don't understand the first point in the contribution on the PLMs geometry-preservation and anisotropy. Isn't this already known?
>
> **Reply-3**: We thank the reviewer for this comment. While anisotropy is a known issue, prior studies mainly focus on hidden representations (Ethayarajh, 2019) or word-level embeddings (Gao et al., 2019) after pre-training. Our work specifically targets input token-level embeddings of PLMs and, more importantly, how they change during fine-tuning. We will revise our contribution to clarify our two new findings:
> - Building on prior findings of anisotropy in hidden representations (Ethayarajh, 2019; Cai et al., 2021) and in word-level embeddings (Gao et al., 2019), our contributions are as follows.
>     - Insights into PLMs' Token Embedding Degeneration. We first confirm that the input token embeddings of encoder-based PLMs are also anisotropic, with rare tokens suffer from semantic unrelatedness. Based on this, we provide two new observations regarding fine-tuning dynamics: (i) The relative geometry of token embeddings is largely preserved during fine-tuning, meaning that fine-tuning does not fix the semantic unrelatedness for rare tokens. (ii) Directly improving isotropy does not help recover semantic meaning. Instead, it introduces geometric fragility during fine-tuning.
>
> **References**:
>
> [1]. Ethayarajh Kawin. How Contextual are Contextualized Word Representations? Comparing the Geometry of BERT, ELMo, and GPT-2 Embeddings. EMNLP, 2019.
>
> [2]. Jun Gao, Di He, Xu Tan, Tao Qin, Liwei Wang, Tie-Yan Liu. Representation Degeneration Problem in Training Natural Language Generation Models. ICLR, 2019.
>
> [3]. Xingyu Cai and Jiaji Huang and Yuchen Bian and Kenneth Church. Isotropy in the Contextual Embedding Space: Clusters and Manifolds. ICLR, 2021.

---

> ### Author Response · Authors · 2026-05-09
> **Part-II: Reply to Reviewer FzmM**
>
> **Question-4**: Using methods like sentenceBert through contrastive learning, don't we end up with semantically meaningful embeddings?
>
> **Reply-4**: We thank the reviewer for this question. The reviewer is correct that methods such as SentenceBERT can use a definition sentence as input to produce a semantically meaningful vector. However, DefinitionEMB is distinguished by two key differences:
>
> - **First, the granularity of output vectors is different**. SentenceBERT is designed to produce a single vector for an entire input sequence. In contrast, modern PLMs use subword vocabularies where most subwords lack individual definitions, making it impossible to obtain their embeddings directly from a dictionary. To bridge this gap, our DefinitionEMB uses the full-word definition as input and decomposes its semantic information into multiple embeddings, one for each constituent subword. This ensures the output matches the architectural requirements of the target PLM.
>
> - **Second, the resulting vector reside in different representation space**. SentenceBERT creates vectors in its own unique representation space, which has a different geometric distribution than the PLMs’ embedding space, like RoBERTa or BART. To replace an embedding within a pre-trained PLM, the new vector must be compatible with the target model's existing space. This ensures that the subsequent layers (like attention and layer normalization) can process the new vector correctly using their pretrained parameters. DefinitionEMB ensures this compatibility by using a "mimicking objective." This constrains the new embeddings to match the geometric properties of the target model’s original space. SentenceBERT, by design, has no such constraint because its vectors are not meant to be plugged back into another model’s input layer.
>
> **We have revised our paper to help audience understand the difference in output granularity as follows**:
> - “SentenceBERT (Reimers & Gurevych, 2019) is designed to generate a single
> vector for an entire sentence. In contrast, modern PLMs use subword vocabularies where most subwords lack individual definitions, making it impossible to obtain their embeddings directly from a dictionary. To bridge this gap, our DefinitionEMB uses the full-word definition as input and decomposes its semantic information into multiple embeddings, one for each constituent subword. This ensures that the generated embeddings are consistent with the PLM's subword-level architecture.”
>
>
> **Reference**:
>
> [1]. Reimers Nils and Gurevych Iryna. Sentence-BERT: Sentence Embeddings using Siamese BERT-Networks. EMNLP, 2019.